# Future nitrogen availability and its effect on carbon sequestration in Northern Eurasia

David W. Kicklighter [1], Jerry M. Melillo[1], Erwan Monier [2,3], Andrei P. Sokolov [2] & Qianlai Zhuang [4]

Nitrogen (N) availability exerts strong control on carbon storage in the forests of Northern Eurasia. Here, using a process-based model, we explore how three factors that alter N availability—permafrost degradation, atmospheric N deposition, and the abandonment of agricultural land to forest regrowth (land-use legacy)—affect carbon storage in the region's forest vegetation over the 21st century within the context of two IPCC global-change scenarios (RCPs 4.5 and 8.5). For RCP4.5, enhanced N availability results in increased tree carbon storage of 27.8 Pg C, with land-use legacy being the most important factor. For RCP8.5, enhanced N availability results in increased carbon storage in trees of 13.4 Pg C, with permafrost degradation being the most important factor. Our analysis reveals complex spatial and temporal patterns of regional carbon storage. This study underscores the importance of considering carbon-nitrogen interactions when assessing regional and sub-regional impacts of global change policies.

[1] The Ecosystems Center, Marine Biological Laboratory, 7 MBL St., Woods Hole, MA 02543, USA. [2] Joint Program on the Science and Policy of Global Change, Massachusetts Institute of Technology, 77 Massachusetts Ave., Cambridge, MA 02139, USA. [3] Department of Land, Air and Water Resources, University of California-Davis, 247 Hoagland Hall, Davis, CA 95616, USA. [4] Department of Earth, Atmospheric, and Planetary Sciences and Department of Agronomy, Purdue University, 550 Stadium Mall Drive, West Lafayette, IN 47907, USA. Correspondence and requests for materials should be addressed to D.W.K. (email: dkicklighter@mbl.edu)

The availability of soil inorganic nitrogen (N) is a critical controller of plant productivity and carbon (C) sequestration in many temperate and boreal ecosystems[1–4] including those in Northern Eurasia[5]. Human activities have directly altered N availability in these ecosystems by providing N subsidies associated with enhanced atmospheric N deposition from fossil fuel combustion, and the application of N fertilizers to croplands[6]. These activities have also altered environmental conditions, including climate, to indirectly affect the metabolic rates of biological processes, such as microbially mediated transformations of organic N compounds to inorganic N (i.e., N mineralization) associated with the decomposition of soil organic matter (SOM) and biological N fixation, to potentially increase N availability[6–12]. Warming-induced permafrost degradation also provides an additional N subsidy to vegetation.

Historically, in high latitude ecosystems such as those in Northern Eurasia, permafrost has protected some ancient SOM from decomposition and the associated cool soil thermal regimes have led to slow decay rates of the more recent SOM to limit N availability to vegetation[9–11]. With warming, more soil N may become available to vegetation, if favorable moisture conditions exist, as SOM decay rates increase in the thawed soil layers[9–12]. In addition, permafrost degradation will expose some protected SOM to decomposition[13,14] and increase N mineralization to provide an additional recycled N subsidy to vegetation that is not currently available. Because this recycled N subsidy is associated with a concurrent loss of C from the enhanced decomposition of SOM, the potential effect of this enhanced N availability on net land C sequestration will depend on the type of vegetation cover. As the carbon-to-nitrogen ratio (C:N) of wood is an order of magnitude greater than the C:N of SOM[15], trees may be able to sequester more atmospheric carbon dioxide from the recycled N subsidy than is lost from SOM decomposition associated with permafrost degradation. In contrast, the C:N of non-woody vegetation is more similar to the C:N of SOM such that the benefits of this recycled N subsidy on C sequestration may be more limited.

Besides warming, land-use legacies may also provide a recycled N subsidy to influence future land C sequestration dynamics. The abandonment of agricultural land (i.e., croplands and pastures) allows access to N sources not currently available to the subsequent natural ecosystem. This N subsidy to natural ecosystems is balanced by the exact corresponding loss of N from agricultural lands at the regional scale and changes over time based on the area of agricultural land abandoned. As a legacy of past land management, this N subsidy may be enhanced from past fertilizer applications to croplands[16,17]. Although fertilizers are applied to croplands to enhance crop yield, some of the fertilizer N will remain as part of the crop residues that eventually become part of the SOM. With the abandonment of croplands, decomposition and mineralization of this legacy fertilizer-enhanced SOM can then increase N availability to the subsequent natural vegetation to influence land C sequestration. Similar to permafrost degradation, the influence of this legacy fertilizer N subsidy on land C sink/source dynamics will depend on the cover type of the secondary vegetation.

Because the N subsidies from permafrost degradation and land-use legacies depend on the recycling of N from SOM, the potential benefits of these N subsidies need to be evaluated within the context of other environmental conditions that affect SOM decomposition and mineralization, or that provide other N subsidies such as atmospheric N deposition[18–22]. In addition, the benefits of these recycled N subsidies to future land C sequestration will lag an initial loss of C associated with the enhanced decomposition associated with exposure of new SOM. This suggests that the response of land C sequestration to climate and land-use change may evolve over time in Northern Eurasian ecosystems. While previous modeling studies[23–26], have shown the importance of carbon-nitrogen interactions on the overall response of land C sequestration in pan-arctic ecosystems to future warming, they have not examined how changes in climate, land use, and atmospheric chemistry may interact to affect the influence of future N availability on C sequestration in these ecosystems.

To assess the consequences of N subsidies from permafrost degradation, land-use legacies, and atmospheric N deposition on N availability and land C sequestration in Northern Eurasia during the 21st century, we use a process-based biogeochemistry model, the Terrestrial Ecosystem Model (TEM, see Methods) with a rising greenhouse gas emissions scenario (Representative Concentration Pathway or RCP8.5) and a climate stabilization scenario (RCP4.5). Besides TEM accounting for the effects of atmospheric N deposition, biological N fixation, and carbon–nitrogen interactions on land C dynamics, including adjustments imposed by a changing soil thermal regime[24], our dynamic cohort approach of representing land-use change (see Methods) allows us to track, in detail, the consequences of both past and future land management decisions on future land C storage and ecosystem productivity[27,28].

Our results show that the enhanced C sequestration by trees associated with N subsidies from permafrost degradation, land-use legacies and atmospheric N deposition (13.4 Pg C under RCP8.5, 27.8 Pg C under RCP4.5) account for 33% (RCP8.5) to 52% (RCP4.5) of the net C sequestration projected for Northern Eurasia during the 21st century. The relative importance of these N subsidies depends on the climate and land policies being implemented (permafrost degradation for RCP8.5, land-use legacies for RCP4.5). The asynchronous timing of C gained by trees from these N subsidies with the associated C loss from soils causes the size and geographical distribution of important sub-regional C sources and sinks to evolve over time. Permafrost degradation tends to enhance C sequestration during the latter part of the century such that forests currently underlain by permafrost account for about twice the current share of the annual forest C sink in Northern Eurasia by the end of the 21st century. In contrast, the relative benefits of the enhanced C sinks from the abandonment of agricultural land tend to diminish over time with forest regrowth. Thus, consideration of carbon-nitrogen interactions is important when assessing sub-regional and regional impacts of global change (i.e., climate change plus land-use change).

## Results

**Current C stocks and C sinks in Northern Eurasia.** Our estimate of the amount of carbon stored in SOM (374 Pg C) in Northern Eurasia is similar to previous inventory estimates[29–34]. Our estimate of vegetation carbon (109 Pg C) is higher than recent estimates (Supplementary Table 1) at least in part because we have not included wildfire effects in our simulations[27,35,36]. However, the bias might also be related to how well our spatially explicit time-series data set represents historical land-use change[28]. Our estimate of current annual carbon sequestration (0.34 Pg C year$^{-1}$) falls in the middle of several other estimates[34,36–39] for Northern Eurasia (Supplementary Table 2). This estimate is on the higher end of estimates[37] (0.02 to 0.35 Pg C year$^{-1}$) by dynamic global vegetation models, a number of which do not consider the influence of N availability on land C dynamics[40] and tend to have a simple representation of soil thermal effects on land C (and N) dynamics[41]. In contrast, our C sequestration estimate is on the lower end of estimates developed using other approaches such as inventory methods[34,36–38] (0.24

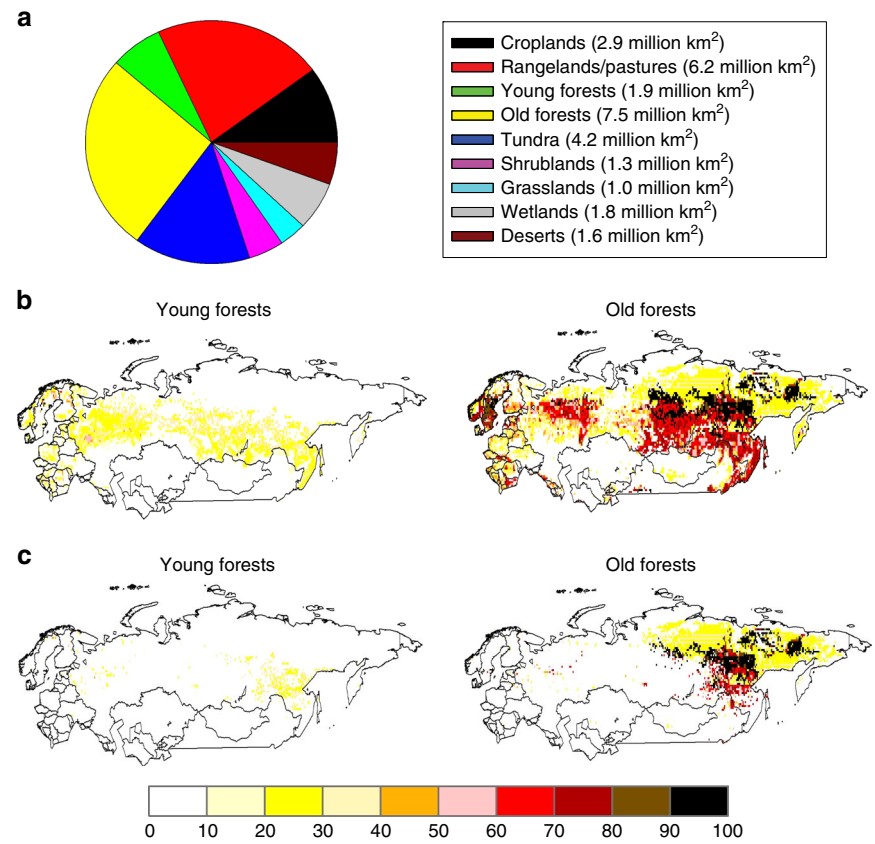

**Fig. 1** Land cover characteristics of Northern Eurasia. **a** Pie chart of the relative areas of land cover in the study area during the year 2000. **b** Percent cover of all young (stand age <120-years-old) and old (stand age at least 120-years-old) forests during the year 2000. **c** Percent cover of young and old forests underlain by permafrost during the year 2000. The spatial distributions of other land cover types represented in the pie chart are given in Supplementary Fig. 2a

to 0.76 Pg C year$^{-1}$), eddy covariance[37] (0.34 to 0.69 Pg C year$^{-1}$), and inverse modeling[37,39] (0.35 to 1.35 Pg C year$^{-1}$), even when the inverse calculations are constrained by Siberian field data[39] (0.35 to 0.56 Pg C year$^{-1}$).

Similar to these other studies, we estimate that forests account for most of the current C sequestration (0.28 Pg C year$^{-1}$) in the region. In 2000, we estimate that forests cover 33% of Northern Eurasia (Fig. 1a, b, Supplementary Table 3) and that 29% of all forests in the region were underlain by permafrost (Fig. 1c, Supplementary Table 4, Supplementary Fig. 1). Our analyses indicate that these permafrost forests accounted for only about 10% of the region's C sequestration at the beginning of the 21st century.

Besides our C estimates, our simulated mean estimate of 16 Tg N of fertilizers being applied to croplands in Northern Eurasia during the 1990s is similar to the estimate of 15 Tg N estimated by Galloway et al.[6] for the region covered by Europe/Former Soviet Union for the same time period and the 15.9 Tg N estimated by Matthews[42] for the former Soviet Union plus Scandinavia during the years 1984/1985.

**Future C sinks and N availability in Northern Eurasia**. The RCP scenarios[43,44] cover many aspects of global change including changes in climate, atmospheric chemistry, and land use (Fig. 2). In the RCP8.5 scenario, most of the land-use changes in Northern Eurasia are related to timber harvests with some displacement and abandonment of agricultural land (Fig. 2c, Supplementary Fig. 2a, b). In contrast, the abandonment of agricultural land accounts for a larger proportion of the land-use change in the

RCP4.5 scenario, with less timber harvests occurring (Fig. 2c, Supplementary Fig. 2a, c). This abandonment of agricultural land results in a 12% increase of forest area over the 21st century in the RCP4.5 scenario (Supplementary Table 3).

Over the 21st century, we project Northern Eurasian ecosystems will sequester 40.6 Pg C under the RCP8.5 scenario and 53.7 Pg C under the RCP4.5 scenario (Supplementary Table 5). In the RCP8.5 scenario, about one-third of the C assimilated by vegetation is compensated by concurrent C losses from soils (Fig. 3, Supplementary Tables 6 and 7). The losses are caused by enhanced SOM decomposition associated with warmer soil temperatures (Supplementary Fig. 3) and permafrost degradation (Supplementary Fig. 1). The less intense warming under the RCP4.5 scenario causes less C to be assimilated by vegetation and less C to be lost from soils such that soil losses offset only 2% of the C assimilated by vegetation and more C is sequestered overall by Northern Eurasian ecosystems in this scenario (Fig. 3).

For both scenarios, forest ecosystems, particularly boreal forests, function as the region's dominant net carbon sink during the 21st century (50.2 Pg C under RCP4.5, 43.2 Pg C under RCP8.5; Fig. 3, Supplementary Table 5). Under the milder RCP4.5 scenario, more C is sequestered by young forests (stands less than 120-years-old) than old forests, whereas more C is sequestered by old forests than young forests under the warmer and wetter RCP8.5 scenario. Because forests dominate both the magnitude and temporal trends in C sequestration and have a large influence on N availability in the region throughout the 21st century (Fig. 4), we examine the factors influencing N availability in forests.

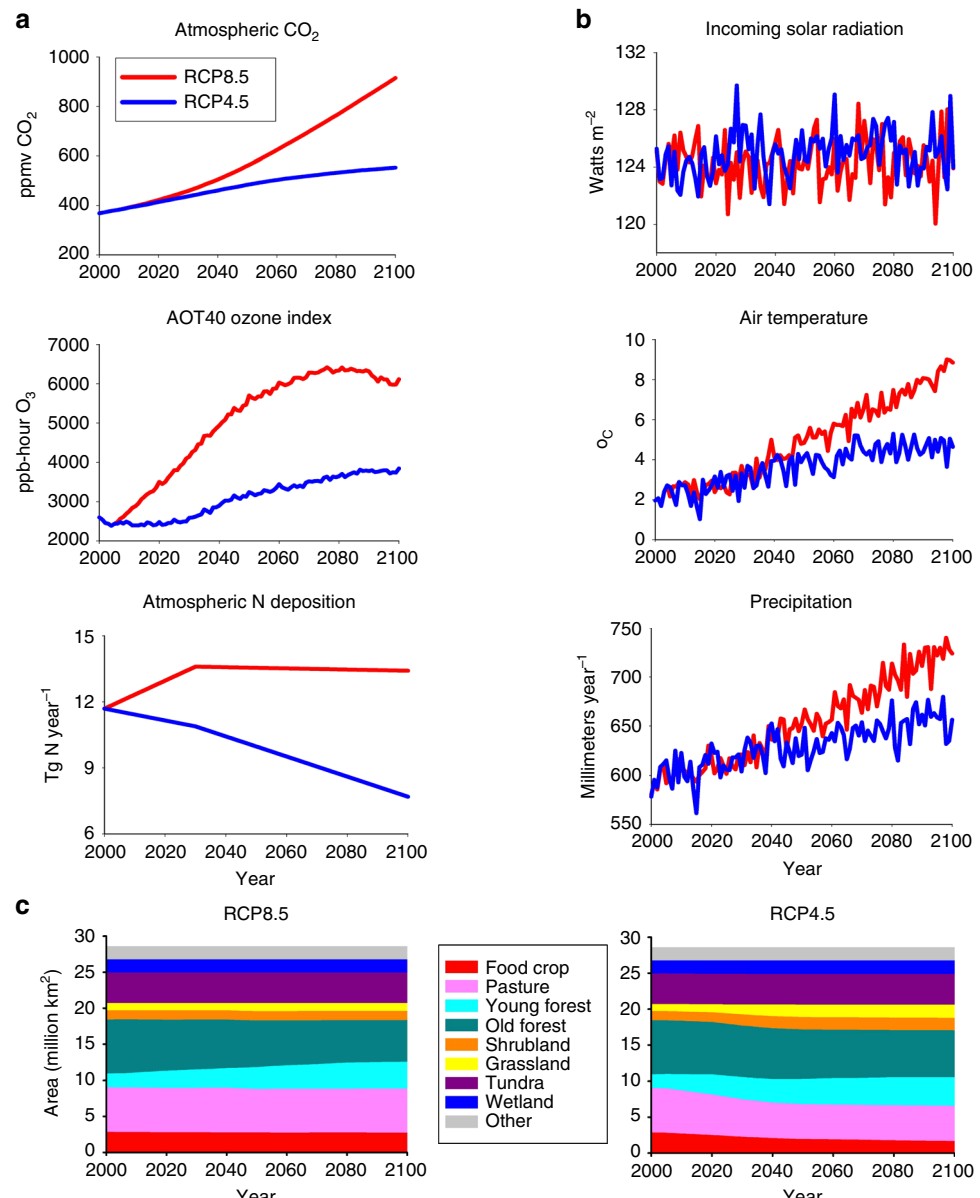

**Fig. 2** Regional characteristics of projected global change by the RCP8.5 and RCP4.5 scenarios during the 21st century for Northern Eurasia. **a** Atmospheric chemistry as represented by mean annual atmospheric $CO_2$ concentrations, mean annual AOT40 ozone index, and annual atmospheric N deposition. **b** Climate as represented by mean annual incoming solar radiation, mean annual air temperature, and annual precipitation. **c** Land-cover change

In Northern Eurasia over the 21st century, slightly more N is available to forests overall under the RCP4.5 scenario than the RCP8.5 scenario (Fig. 3a, Supplementary Table 8). Although more N is available to old forests than young forests in both scenarios, the relative amount available to young forests compared to old forests is larger under the RCP4.5 scenario. Most of this forest available N (85–88%) is recycled N, produced from N mineralization associated with the decomposition of plant litter and soil organic matter (Supplementary Tables 9–12). In addition, the temporal changes in N availability over the 21st century under both scenarios (Fig. 4b) are mostly influenced by the temporal changes in net N mineralization. Therefore, we focus our analyses on how permafrost degradation, land-use legacies, and atmospheric N deposition influence net N mineralization and C sequestration in Northern Eurasian forests.

**Impact of N subsidies on forest net N mineralization**. Overall, permafrost degradation, land-use legacies, and atmospheric N

deposition enhance net N mineralization in forests by between 406 Tg N (RCP8.5) and 1036 Tg N (RCP4.5) and account for between 8.1 and 19.8% of forest net N mineralization over the 21st century (Table 1). For the RCP8.5 scenario, the N subsidies associated with enhanced net N mineralization from permafrost degradation, land-use legacy, and atmospheric N deposition (Table 1, Supplementary Tables 9, 13–18) are about the same magnitude as the N inputs from atmospheric N deposition to forests over the 21st century (417 Tg N), whereas these N subsidies are almost triple the N inputs from atmospheric N deposition (353 Tg N) for the RCP4.5 scenario. The enhanced net N mineralization from atmospheric N deposition is a legacy from past atmospheric N deposition, where the N has already been incorporated into organic matter by vegetation or microbes and is then made available again from the remineralization of the resulting SOM. This legacy input of N occurs in addition to the direct inputs of N from atmospheric N deposition to forests during the 21st century to represent the total N subsidies to forests from atmospheric N deposition.

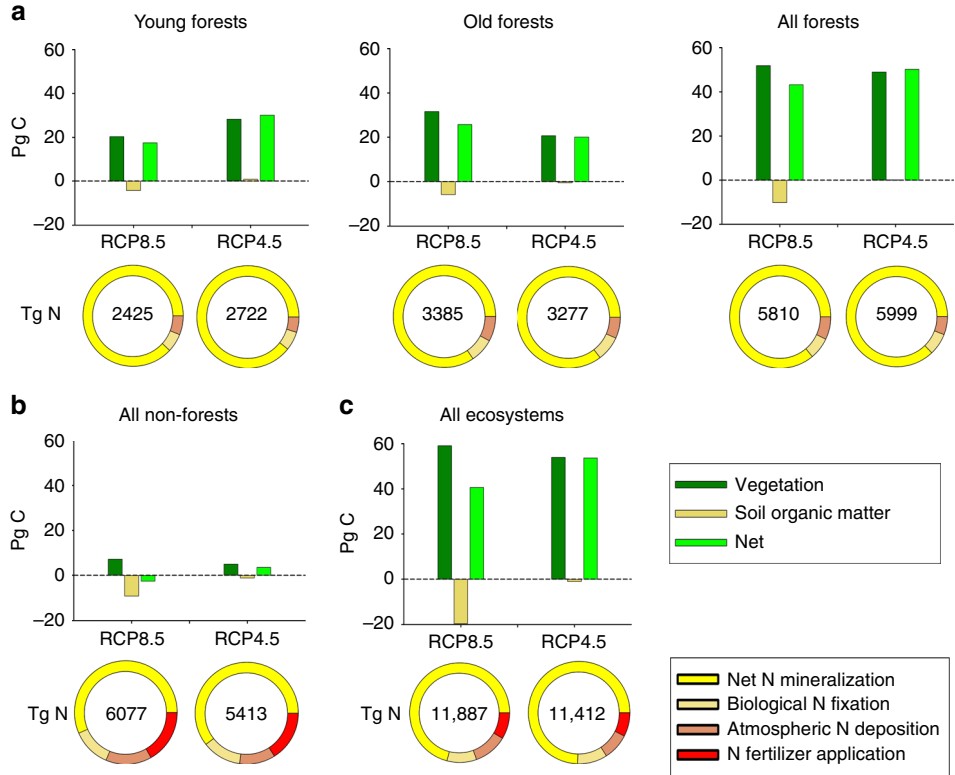

**Fig. 3** Characteristics of future C sequestration and N availability for Northern Eurasian ecosystems under the RCP8.5 and RCP4.5 scenarios. **a** Forest ecosystems include young forests (stand age <120-years-old) and old forests (stand age at least 120-years-old). **b** Non-forested ecosystems (combination of tundra, grasslands, shrublands, deserts, wetlands, croplands, and pastures). **c** All ecosystems. Positive values in the graphs represent cumulative C sequestration over the 21st century and negative values represent C losses. Circles represent the partitioning of N availability among net N mineralization, biological N fixation, atmospheric N deposition and N fertilizer application. Values within circles represent total N availability over the 21st century

Our analysis indicates that fertilizer applications to croplands under the RCP4.5 scenario account for 41.1% of the N subsidy to forests from cropland abandonment with about a third of the fertilizer subsidy (222 Tg N) derived from fertilizers applied to croplands between 1950 and 2000. Under the RCP8.5 scenario, fertilizer applications to croplands account for 43.9% of the legacy N subsidy from croplands, with over two-thirds of the fertilizer subsidy (25 Tg N) derived from fertilizers applied before year 2001.

**Impact of N subsidies on forest C sequestration**. The N subsidies from permafrost degradation, land-use legacies and atmospheric N deposition allow forest vegetation to sequester between 13.4 Pg C (RCP8.5) and 27.8 Pg C (RCP4.5), which account for 25.8% (RCP8.5) to 56.7% (RCP4.5) of the total C sequestered by forest vegetation (Table 1). In both scenarios, the N subsidies from permafrost degradation allow more C to be sequestered by vegetation than the N subsidies from atmospheric N deposition, including the effects of direct inputs from atmospheric N deposition (Supplementary Tables 6, 19–24). In contrast, N subsidies from land-use legacies also allow more C to be sequestered by forest vegetation (21.5 Pg C) than N subsidies from permafrost degradation under the RCP4.5 scenario, but less C to be sequestered by forest vegetation (3.1 Pg C) under the RCP8.5 scenario than either permafrost degradation or atmospheric N deposition. This difference is mostly a result of forest regrowth on large areas of croplands (72 million ha, Supplementary Table 25) and pastures (47 million ha, Supplementary Table 26) that have been abandoned under the RCP4.5 scenario compared to the RCP8.5 scenario (8 million ha of croplands, 11

million ha of pastures). While legacy effects of fertilizer application to croplands account for 10.8% of the total amount of C sequestered by forest vegetation regrowing on abandoned croplands during the 21st century under the RCP4.5 scenario, these legacy fertilizer effects account for only 2.9% of C sequestered by all forest vegetation in this scenario (Table 1, Supplementary Tables 6, 20–22). Finally, the benefits of the N subsidies on C sequestration differed by forest age. The old forests received larger benefits than young forests from the N subsidies associated with permafrost degradation and atmospheric N deposition. In contrast, the young forests received benefits from the N subsidies associated with land-use legacies while the old forests received none.

While permafrost degradation and land-use legacy effects provide increased N availability to enhance C sequestration by vegetation, these N subsidies comes at a cost of concurrent losses in soil organic C (Table 1, Supplementary Tables 7, 27–32) caused by the decomposition/mineralization of SOM to generate the inorganic N used by vegetation. For permafrost degradation, the gain of vegetation C is only 29.8% (RCP4.5) to 49.2% (RCP8.5) of the loss of soil organic C such that permafrost degradation overall diminishes C sequestration in these forests in both scenarios (Table 1, Supplementary Tables 5 and 33). The losses depend on climate and land use so that the relative gains in vegetation C compared to soil C losses of old forests are similar for the two scenarios (30.8% for RCP4.5; 35.5% for RCP8.5), but are very different for the young forests with a net loss of C under the cooler RCP4.5 scenario and a net gain of C under the warmer RCP8.5 scenario.

For legacy fertilizer effects, the gain in vegetation C is only 50% of the loss of soil C in both scenarios indicating that legacy

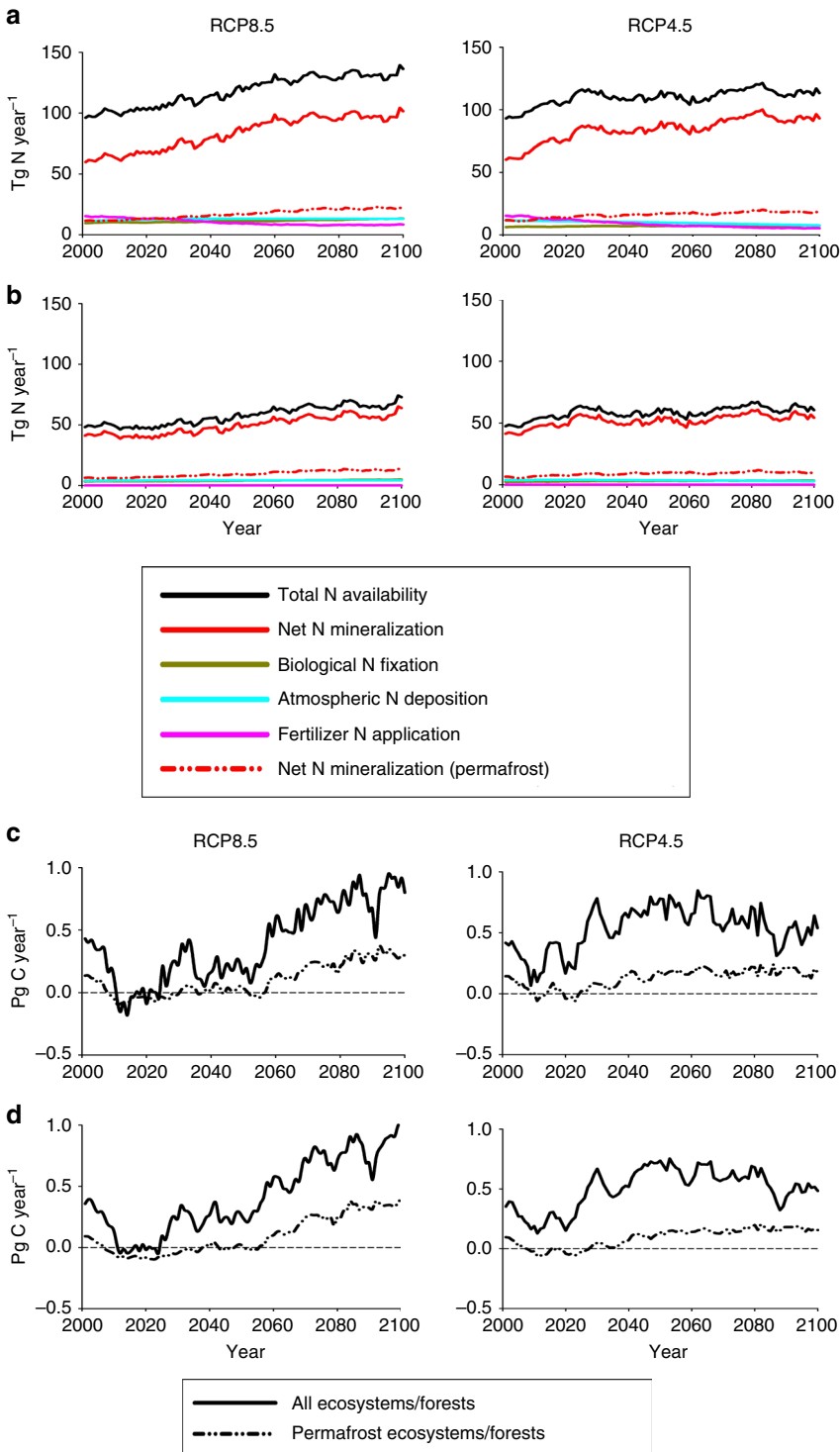

**Fig. 4** Temporal trends in annual N availability and annual net C sequestration in Northern Eurasia projected by TEM based on the RCP8.5 and RCP4.5 scenarios during the 21st century. **a** Trends in total N availability are partitioned by the contributions of net N mineralization, biological N fixation, atmospheric N deposition, and the application of N fertilizers for all ecosystems. **b** Same as **a**, but for all forests. The contribution of net N mineralization for ecosystems and forests underlain by permafrost in 2000 are also shown in **a** and **b**, respectively. **c** Trends in net C sequestration for all ecosystems. **d** Same as **c**, but for all forests. Net C sequestration in ecosystems and forests underlain by permafrost in 2000 are also shown in **c** and **d**, respectively

fertilizer effects overall also tend to diminish C sequestration in forests in these scenarios (Table 1, Supplementary Tables 34-36). The fertilizer-induced loss of soil organic C in our simulations is because fertilizers increase the amount of C stored in SOM when croplands are abandoned to forests and enhance heterotrophic respiration. For the RCP4.5 scenario, fertilizer applications

increased the amount of soil organic C abandoned to forests by 3.0 Pg C (Fig. 5g). Interestingly, most of the additional soil organic carbon (2.4 Pg C) is a result of fertilizer applications to croplands after the year 2000 rather than the pre-2001 applications (0.6 Pg C), indicating that crops are responding to more favorable conditions during the early part of the 21st

**Table 1 Effect of N subsidies on cumulative net N mineralization (NetNMin), and C sequestration in forest vegetation (VegC_NEW) and soils (TotSOC_NEW) over the 21st century**

| Factor | RCP8.5 | | | RCP4.5 | | |
|---|---|---|---|---|---|---|
| | NetNMin (Tg N) | VegC$_{NEW}$ (Pg C) | TotSOC$_{NEW}$ (Pg C) | NetNMin (Tg N) | VegC$_{NEW}$ (Pg C) | TotSOC$_{NEW}$ (Pg C) |
| Permafrost Degradation[a] | | | | | | |
| Young forests | 84 | 2.3 | −1.6 | 32 | 0.3 | −1.4 |
| Old forests | 110 | 3.9 | −11.0 | 112 | 3.3 | −10.7 |
| All forests | 194 | 6.2 | −12.6 | 144 | 3.6 | −12.1 |
| Cropland Abandonment[b] | | | | | | |
| Unfertilized | | | | | | |
| Young forests | 32 | 1.2 | 0.3 | 318 | 11.6 | 2.3 |
| Old forests | 0 | 0.0 | 0.0 | 0 | 0.0 | 0.0 |
| All forests | 32 | 1.2 | 0.3 | 318 | 11.6 | 2.3 |
| Pre-2001 | | | | | | |
| Young forests | 17 | 0.1 | −0.2 | 77 | 0.9 | −0.5 |
| Old forests | 0 | 0.0 | 0.0 | 0 | 0.0 | 0.0 |
| All forests | 17 | 0.1 | −0.2 | 77 | 0.9 | −0.5 |
| Post-2000 | | | | | | |
| Young forests | 8 | 0.0 | 0.0 | 145 | 0.5 | −2.3 |
| Old forests | 0 | 0.0 | 0.0 | 0 | 0.0 | 0.0 |
| All forests | 8 | 0.0 | 0.0 | 145 | 0.5 | −2.3 |
| Total | | | | | | |
| Young forests | 57 | 1.3 | 0.1 | 540 | 13.0 | −0.5 |
| Old forests | 0 | 0.0 | 0.0 | 0 | 0.0 | 0.0 |
| All forests | 57 | 1.3 | 0.1 | 540 | 13.0 | −0.5 |
| Pasture Abandonment[b] | | | | | | |
| Young forests | 41 | 1.8 | 0.4 | 235 | 8.5 | 1.8 |
| Old forests | 0 | 0.0 | 0 | 0 | 0.0 | 0.0 |
| All forests | 41 | 1.8 | 0.4 | 235 | 8.5 | 1.8 |
| Atmospheric N Deposition[c,d] | | | | | | |
| Young forests | 16 | 1.2 | 0.2 | 28 | 0.8 | 0.3 |
| Old forests | 98 | 2.9 | 1.3 | 89 | 1.9 | 1.1 |
| All forests | 114 | 4.1 | 1.5 | 117 | 2.7 | 1.4 |
| Total N Subsidies[d] | | | | | | |
| Young forests | 198 | 6.6 | −0.9 | 835 | 22.6 | 0.2 |
| Old forests | 208 | 6.8 | −9.7 | 201 | 5.2 | −9.6 |
| All forests | 406 | 13.4 | −10.6 | 1036 | 27.8 | −9.4 |
| Enhanced Metabolism[e] | | | | | | |
| Young forests | 1935 | 13.7 | −3.4 | 1593 | 5.7 | 0.6 |
| Old forests | 2650 | 24.8 | 3.8 | 2598 | 15.5 | 9.0 |
| All forests | 4585 | 38.5 | 0.4 | 4191 | 21.2 | 9.6 |
| Total | | | | | | |
| Young forests | 2133 | 20.3 | −4.3 | 2428 | 28.3 | 0.8 |
| Old forests | 2858 | 31.6 | −5.9 | 2799 | 20.7 | −0.6 |
| All forests | 4991 | 51.9 | −10.2 | 5227 | 49.0 | 0.2 |

[a]Responses based on the differences in TEM estimates described in the Permafrost degradation section of Methods
[b]Responses based on the differences in TEM estimates described in the Land-use legacies including fertilizer applications section of Methods
[c]Responses based on the differences in TEM estimates described in the Atmospheric N deposition section of Methods.
[d]Estimates for VegC$_{NEW}$ and TotSOC$_{NEW}$ also include the effects of the direct inputs from atmospheric N deposition in addition to the N deposition-enhanced net N mineralization
[e]Estimates include the effects of climate change and land-use change on N recycling from enhanced metabolism of vegetation and microbes

century. We also estimate that fertilizers increase the amount of soil organic N abandoned to forest by 364 Tg N with 232 Tg N as a result of fertilizers being applied after year 2000 and 132 Tg N from the pre-2001 fertilizer applications (Fig. 5h).

Overall, the net loss of C associated with permafrost degradation and legacy fertilizer effects is more than compensated by the effects of atmospheric N deposition (Table 1, Supplementary Tables 5 and 37) and the regrowth of forest vegetation on abandoned pastures (Table 1, Supplementary Table 38) and the unfertilized component of abandoned croplands in both scenarios (Table 1, Supplementary Tables 35 and 36). This result occurs,

however, because the net gain of C in young forests from all N subsidies overcome the concurrent net loss of C in old forests associated with these N subsidies. It is only after the influence of climate on N recycling from enhanced vegetation and microbial metabolism is considered that these old forests accumulate enough vegetation biomass and SOM to become C sinks (Table 1).

**Evolution of N subsidy impacts on forest C sequestration.** While both legacy fertilizer effects and permafrost degradation tend to diminish carbon sequestration in Northern Eurasian

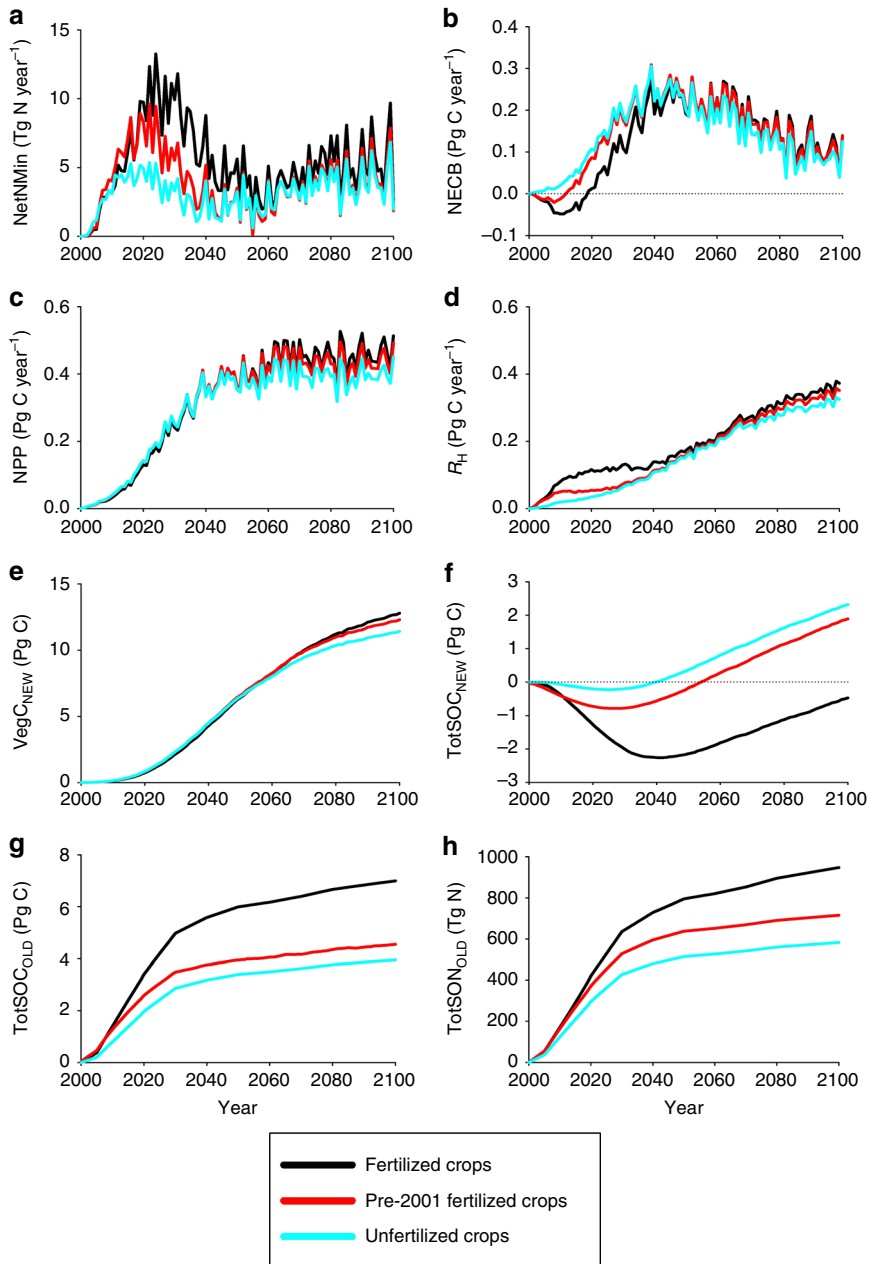

**Fig. 5** Influence of N fertilizer applications on temporal trends in C and N fluxes during forest regrowth on abandoned land that was covered by croplands during the year 2000. **a** Annual net N mineralization (NetNMin). **b** Annual C sequestration/loss as represented by net ecosystem carbon balance (NECB). **c** Annual net primary production (NPP). **d** Annual heterotrophic respiration ($R_H$). **e** Accumulation of new-carbon sequestered by forest vegetation (VegC_NEW). **f** Accumulation/loss of new-carbon sequestered into forest soil organic matter (TotSOC_NEW). **g** Accumulation of old-soil organic carbon stocks (TotSOC_OLD) as croplands are abandoned to forests. **h** Accumulation of old-soil organic nitrogen stocks (TotSON_OLD) as croplands are abandoned to forests. Simulations include abandoned croplands where crops are never fertilized (unfertilized crops), crops are fertilized between years 1950 and 2000 (pre-2001 fertilized crops), and crops are fertilized between years 1950 and 2100 (fertilized crops)

forests overall, the response of forests to these factors changes over the 21st century affecting the timing of forest carbon source/sink activities. For the legacy fertilizer effects, decomposition of the fertilizer-enhanced SOM (Supplementary Tables 39-41) initially increases heterotrophic respiration ($R_H$, Fig. 5d) and net N mineralization (Fig. 5a), but has little effect on net primary production (NPP, Fig. 5c). This causes a loss of soil organic C (Fig. 5f) so the regrowing forests are initially more of a C source (i.e., lower values of NECB in Fig. 5b) with additional fertilizer applications to crops. After 2050, the increase in inorganic N from the remineralization of the fertilizer-enhanced SOM begins

to increase NPP (Supplementary Tables 42–44) such that vegetation begins to accumulate more C (Fig. 5e) based on past fertilizer applications to crops. As a result, forests regrowing on abandoned fertilized croplands under the RCP4.5 scenario become more of a C sink during the latter part of the century. A similar dynamic occurs with permafrost degradation (Supplementary Fig. 4), where permafrost degradation first causes enhanced loss of soil organic carbon, but the benefits of the associated enhanced net N mineralization on C sequestration by vegetation does not occur until later. It is also interesting to note that our analyses suggest that if permafrost degradation effects are

not considered, soils under old forests would be considered to be a C sink rather than a C source for both scenarios.

**Impact of land-use history and warming interactions.** Climate, land-use history, and geographic location interact to affect the timing, distribution and magnitude of N availability and C sequestration in forest ecosystems over the 21st century. While some interactions are similar between the two global change scenarios, others are different and they lead to different components of Northern Eurasian forests being important C sinks in the two scenarios. In both scenarios, a similar amount of warming occurs during most of the first half of the 21st century (Fig. 2). This warming trend continues during the second half of the century under the RCP8.5 scenario, but warming levels off under the RCP4.5 scenario.

Land-use history affects both the sensitivity of forests to warming and the synchronization of N availability to C sequestration in these forests. The influence of warming, including the effects of N subsidies from permafrost degradation and atmospheric N deposition, on N availability and C sequestration are observed in the old forests, where no disturbance occurred during the 21st century (Fig. 6d). Under the RCP8.5 scenario, linear regression analyses of NECB against year using SigmaPlot 12.5 (regression coefficient ± s.e.) indicated that significant increases in C sequestration rates ($0.0040 \pm 0.0004$ Pg C year$^{-2}$, $r^2 = 0.47$, $p < 0.0001$, $n = 100$) of the undisturbed old forests are supported by concurrent warming-induced increases in net N mineralization ($0.065 \pm 0.011$ Tg N year$^{-2}$, $r^2 = 0.27$, $p < 0.0001$, $n = 100$) throughout the 21st century as determined by corresponding linear regressions of net N mineralization against year. During the first half of the century under the RCP4.5 scenario, warming also increases the rates of both C sequestration ($0.004 \pm 0.001$ Pg C year$^{-2}$, $r^2 = 0.21$, $p = 0.0008$, $n = 50$) and net N mineralization ($0.092 \pm 0.027$ Tg N year$^{-2}$, $r^2 = 0.18$, $p = 0.0012$, $n = 50$) of the undisturbed old forests. However, no significant trends in C sequestration and net N mineralization rates in old forests are observed during the second half of the century under the RCP4.5 scenario when warming has leveled off.

Timber harvest appears to enhance the sensitivity of N availability and C sequestration in young forests to warming (Fig. 6c). Under the warmer RCP8.5 scenario, changes in net N mineralization rates in young forests ($0.170 \pm 0.014$ Tg N year$^{-2}$, $r^2 = 0.60$, $p < 0.0001$, $n = 100$) are more than double the rates of old forests to support larger increases in C sequestration rates after initial losses of C from these forests at the beginning of the century. As a result, the C sequestration rates of young forests and old forests are about equal at the end of the 21st century even though young forests cover only 64% of the area covered by old forests at this time. Similar to the RCP8.5 scenario, net N mineralization rates in secondary forests recovering from timber harvests under the milder RCP4.5 stabilization scenario increase throughout the 21st century by $0.095 \pm 0.018$ Tg N year$^{-2}$ ($r^2 = 0.21$, $p < 0.0001$, $n = 100$). After decreasing during the first half of the century, C sequestration rates of these secondary forest increase during the second half of the century, but the resulting rates are only about half those of undisturbed old forests at the end of the 21st century under the RCP4.5 scenario (Fig. 6d).

In contrast to timber harvest, the regrowth of secondary forests on abandoned croplands and pastures is relatively insensitive to warming. Net N mineralization and C sequestration do not increase over time in these secondary forests, but instead peak earlier during the century before becoming more diminished as these secondary forests mature (Fig. 6a, b). In addition, N availability associated with net N mineralization is not synchronized with C sequestration in these secondary forests. Because inorganic N needs to be created from SOM mineralization before forest vegetation can benefit from the N subsidy from land-use legacies, the peak in C sequestration lags the peak in net N mineralization in these forests. As described earlier, the legacy of past fertilizer applications to croplands enhance C sequestration in vegetation, but these benefits appear to be initially limited by the influence of forest regrowth dynamics on C sequestration and occur later as these forests mature (Fig. 5e). The N subsidy from legacy fertilizer applications, however, does allow forests regrowing on abandoned fertilized croplands to sequester about four times more C per year than those regrowing on abandoned unfertilized pastures at the end of the 21st century. On a per area basis, our estimate of future C sequestration on abandoned agricultural land (251 g C m$^{-2}$ year$^{-1}$) is comparable to an estimate of current carbon sequestration[45] (245 g C m$^{-2}$ year$^{-1}$). The legacy of past N fertilizer applications causes cropland abandonment to add almost twice as much nitrogen per area abandoned (10.6 g N m$^{-2}$ year$^{-1}$) than pasture abandonment (5.9 g N m$^{-2}$ year$^{-1}$) in our simulations.

The different responses of forests regrowing on abandoned agricultural land versus recovering from timber harvest are a result of differences in the timing of the land-use change and the relative importance of N subsidies from permafrost degradation, land-use legacies and atmospheric N deposition. Most of the forests regrowing on abandoned croplands and pastures under the RCP4.5 scenario is located in eastern Europe/western Russia, southern Russia and northeastern China (Fig. 7). These areas are not underlain by permafrost (Supplementary Fig. 1) so these forests do not benefit from N subsidies associated with permafrost degradation. Instead, C sequestration in these regrowing forests are influenced mostly by local environmental conditions on N recycling and the N subsidies from land-use legacies and atmospheric N deposition (Supplementary Fig. 5). As a result, higher rates of forest net N mineralization and C sequestration (Fig. 8), particularly in vegetation (Supplementary Fig. 6a), occur in areas with more mesic conditions, such as eastern Europe/Western Russia[5], and higher inputs of atmospheric N deposition (Supplementary Fig. 5c). In addition, most of the abandonment of croplands and pastures occurs during the first half of the 21st century (Fig. 7) so the benefits of the land-use legacy N subsidies on C sequestration from forest regrowth in these abandoned agricultural lands continue into the second half of the 21st century, but become diminished (Fig. 8a, b, Supplementary Fig. 6b, c) as these forest trees mature. One exception is northeastern China, where more cropland abandonment occurs during the second half of the 21st century (Fig. 7) and supports a higher rate of C sequestration (Fig. 8a, b, Supplementary Fig. 6b, c) during this period.

In contrast, timber harvests occur in both eastern Europe/western Russia and central Siberia in both scenarios throughout the 21st century (Fig. 7). An analysis of the stand age distribution of forests under the RCP8.5 scenario (Supplementary Fig. 7) indicates that most young stands are created from timber harvest during the first half of the century. However, more timber harvests of permafrost forests occur during the second half of the century (Fig. 7, Supplementary Fig. 8, Supplementary Table 45). Thus, C sequestration from the regrowth of these secondary forests after timber harvest are influenced by both changing local environmental conditions on N recycling and N subsidies from permafrost degradation associated with climate change (Supplementary Tables 46–49). Similar to the current pattern of C sources and sinks[37,38], we estimate that timber harvest generally enhances C sequestration in forests of eastern Europe/western Russia, but mostly enhances C losses in permafrost forests of central Siberia during the first half of the 21st century (Fig. 8a).

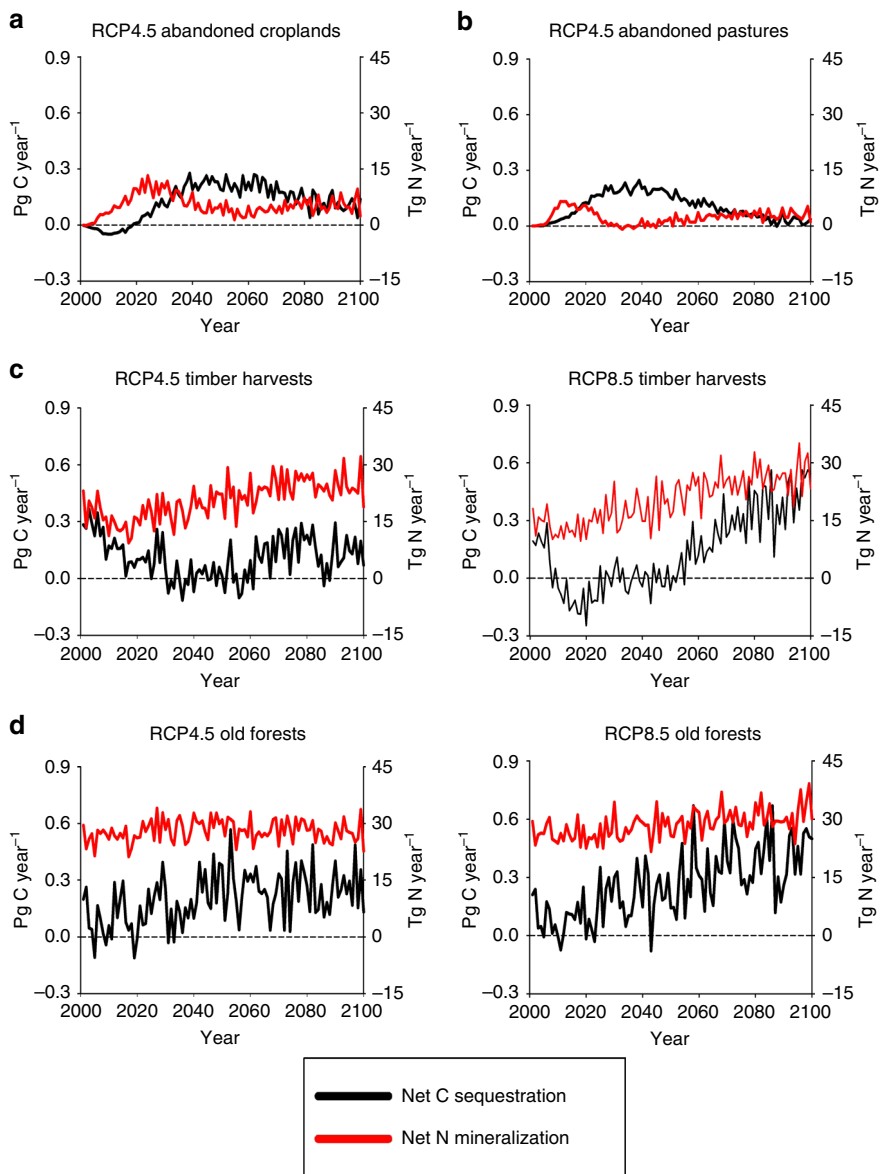

**Fig. 6** Influence of global change on forest C sequestration and net N mineralization. **a** Secondary forests on abandoned croplands under the RCP4.5 scenario. **b** Secondary forests on abandoned pastures under the RCP4.5 scenario. **c** Secondary forests recovering from timber harvest under the RCP4.5 and RCP 8.5 scenarios. **d** Old forests that have not been disturbed by land-use change during the 21st century under the RCP4.5 and RCP8.5 scenarios. Area of secondary forests on abandoned croplands and abandoned pastures are insignificant under the RCP8.5 scenario

Secondary Siberian forests are a C source because the C loss associated with the decomposition of timber slash and SOM, including SOM exposed by permafrost degradation, overwhelms the C assimilated in vegetation biomass (Supplementary Fig. 6b) of the slow growing trees underlain by permafrost during this period. During the second half of the 21st century, however, the trees of these secondary permafrost forests have recovered enough such that more C is being assimilated in vegetation than lost from soils so that these Siberian forests become C sinks (Fig. 8b, Supplementary Fig. 6c).

To better understand this temporal change in forest response dynamics, we conduct a simulation experiment for a larch forest located in central Siberia in which we compare the forest response to a timber harvest that occurs during the year 2030 to the corresponding response to a 2070 timber harvest (see Methods). While a timber harvest generally causes a loss of C from a forest ecosystem, most of the C loss occurs with the removal of tree biomass to create wood products and the burning of fuelwood or

the resulting slash[46]. However, some of the C losses occur over a longer time frame as the decomposition of SOM and the added unburnt slash initially overwhelm the uptake of C by young vegetation. As forests regrow, the uptake of C by vegetation eventually overwhelms the loss of soil C from decomposition such that forests sequester carbon. The timber harvest also disrupts the synchrony of carbon–nitrogen interactions of forest ecosystems that influence the timing of both N availability and the ability to sequester carbon[47]. The addition of slash with a high C:N ratio initially increases N immobilization (i.e., uptake of N by soil microbes) so that increases in net N mineralization (Fig. 9g) associated with the enhanced decomposition are subdued at first. However, as the C:N ratio of the decomposing slash declines with the release of C to the atmosphere, N mineralization of the increasingly labile SOM overwhelms N immobilization to enhance N availability to vegetation. In forests underlain by permafrost, this net N mineralization rate may be further enhanced by the exposure of SOM from permafrost degradation.

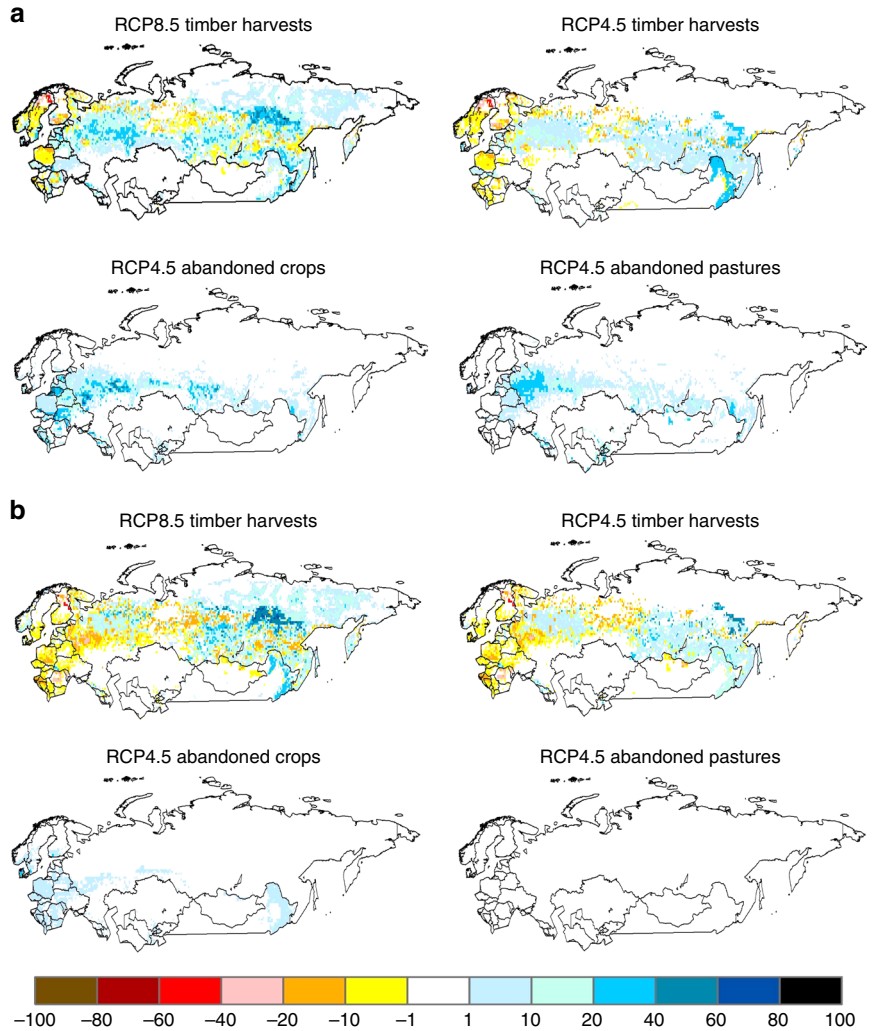

**Fig. 7** Spatial patterns of changes in forest cover as influenced by dominant land management under the RCP8.5 and RCP4.5 scenarios over the 21st century. **a** Changes in percent cover during the first half of the century (2001–2050). **b** Changes in percent cover during the second half of the century (2051–2100). Land management includes timber harvests and the abandonment of pastures and fertilized croplands. Positive values indicate an increase in forest area from cropland or pasture abandonment, or an increase in area of young forests (stand age <120-years-old) from timber harvest. Negative values represent a decrease in area of young forests related to forest aging (i.e., stand age becoming older than 120-years-old) from lack of human disturbance

Eventually, the enhanced net N mineralization rate supports higher NPP that overwhelm the declining heterotrophic respiration such that forest regrowth begins to sequester carbon. As forest recovery continues, the enhanced NPP (Fig. 9d) increases tree biomass (Fig. 9b) to increase litterfall C (Fig. 9e) so that the reduced standing stocks of soil organic C begin to recover (Fig. 9c) and enhance heterotrophic respiration again (Fig. 9f) to reduce the amount of C being sequestered by these forests (Fig. 9a).

The 2070 timber harvest of the larch forest results in a larger initial loss of C than the 2030 timber harvest because the trees have gained more biomass after growing for an additional 40 years. The net loss of C after the 2070 harvest, however, lasts for a shorter period because the warmer conditions (Fig. 9h) enhance the decomposition of the timber slash to decrease the period of elevated heterotrophic respiration and enhances net N mineralization and NPP. In addition to providing a N subsidy to these regrowing forests, the deeper active layer from permafrost degradation diverts less water to neighboring river networks during spring runoff and allows more water to be stored during the growing season[48] to support higher NPP. The increased forest NPP following the 2070 timber harvest then allows both vegetation and soil organic C to recover more rapidly than the 2030 timber harvest. Thus, permafrost degradation and more favorable climate and atmospheric carbon dioxide concentrations (Fig. 2) have sped up the recovery of C storage in secondary permafrost forests after timber harvest to enhance C sequestration. As a result, some permafrost forests experiencing timber harvests later during the 21st century may still be net C sources, but these C losses are being more than compensated by net C sinks by other secondary permafrost forests recovering from earlier timber harvests as supported by enhanced N availability. With the changing responses of permafrost forests to warming and land-use change over the 21st century, permafrost forests are projected to account for a larger and increasing share of annual forest C sequestration in Northern Eurasia during the latter half of the 21st century (Fig. 4d) and account for 19% (RCP4.5) to 24% (RCP8.5) of annual forest C sequestration in 2100.

## Discussion

Our analysis indicates that increases in N availability associated with permafrost degradation, land-use legacies, and atmospheric

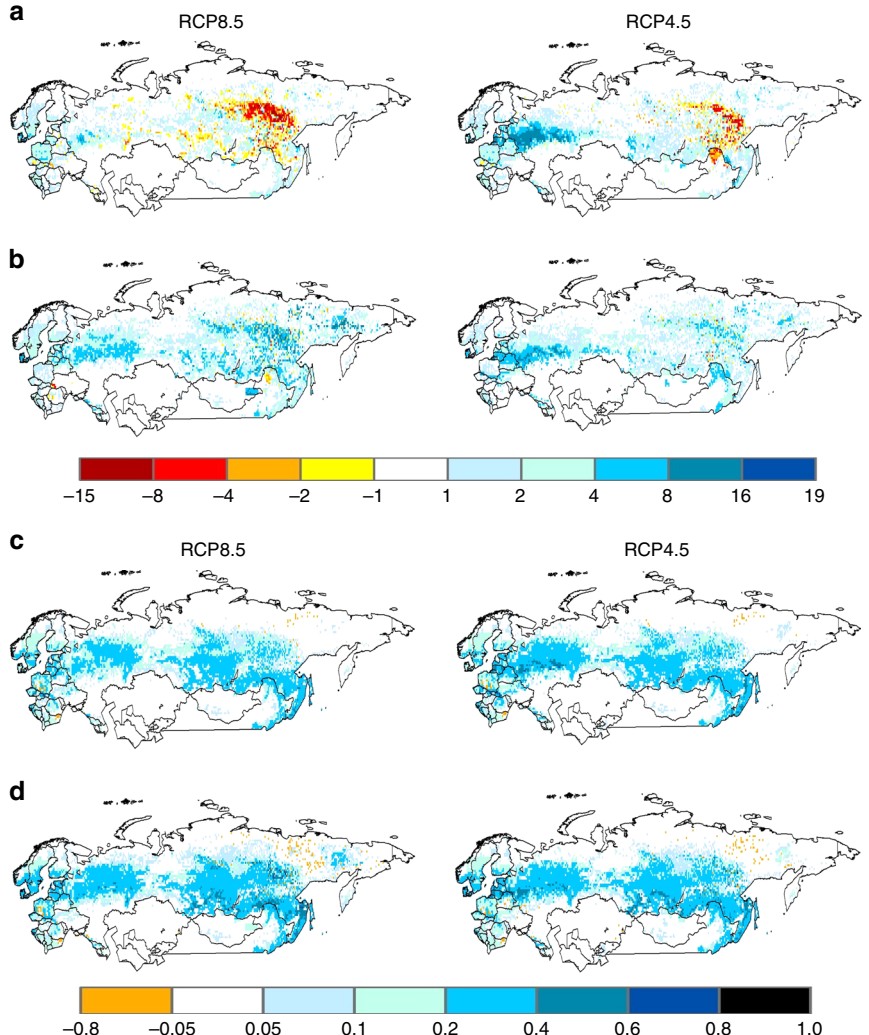

**Fig. 8** Changes in the pattern of net C sequestration (kg C m$^{-2}$) and net N mineralization (kg N m$^{-2}$) in forests across Northern Eurasia during the 21st century under the RCP8.5 and RCP4.5 global change scenarios. **a** Net C sequestration represented by cumulative net ecosystem carbon balance (NECB) during the first half of the century (2001–2050) **b** Same as **a**, but for the second half of the century (2051–2100). **c** Cumulative net N mineralization during the first half of the century (2001–2050). **d** Same as **c**, but for the second half of the century (2051–2100)

N deposition can have a large impact on the magnitude, distribution, and timing of forest C sequestration and projected net C sinks in Northern Eurasia. Overall, the enhanced C sequestration by vegetation from these N subsidies represent between 33.0% (RCP8.5) and 51.8% (RCP4.5) of the net C sinks estimated for Northern Eurasia. The relative importance of these N subsidies on forest C sequestration depends on the type of land-use change that occurs, climate conditions, the presence or absence of permafrost, the amount of atmospheric N deposition, and to a lesser degree, past and future inputs from fertilizer applications, which can all vary significantly among future climate scenarios. Thus, the relative importance of N subsidies under the warmer RCP8.5 scenario (permafrost degradation > atmospheric N deposition > land-use legacy) is different than under the cooler RCP4.5 scenario (land-use legacy >> permafrost degradation > atmospheric N deposition) and also reflects assumed differences in the abandonment of agricultural land and timber harvest between the two scenarios. In addition, the influence of N subsidies varies among different components of Northern Eurasian forests with permafrost degradation and atmospheric N deposition having a large influence on vegetation C sequestration in old forests in both scenarios, but land-use legacies having a large influence on vegetation C sequestration in young forests under only the RCP4.5 scenario. The C gain by forest vegetation from these N subsidies, however, is somewhat dampened by a concurrent C loss from forest soils caused by permafrost degradation and legacies of fertilizer applications to croplands. As a result, we estimate that C sequestration in Northern Eurasia during the 21st century, on average, will likely be similar to current rates of C sequestration. These average rates, however, conceal the influence of important changes in temporal trends and spatial patterns of C sequestration and loss occurring in the region.

The asynchronous timing of C gain by trees and C loss from soils associated with the N subsidies causes the size and geographical distribution of important regional C sources and sinks to evolve over time. Although permafrost degradation tends to diminish C sequestration in forests overall during the 21st century, N subsidies from permafrost degradation help Siberian forests recover from timber harvest more rapidly such that these forests become larger C sinks rather than C sources during the latter part of the century. In contrast, N subsidies from the abandonment of agricultural land, particularly in the western part of the region, during the first half of the 21st century, tend to enhance C sequestration in these forests throughout the 21st

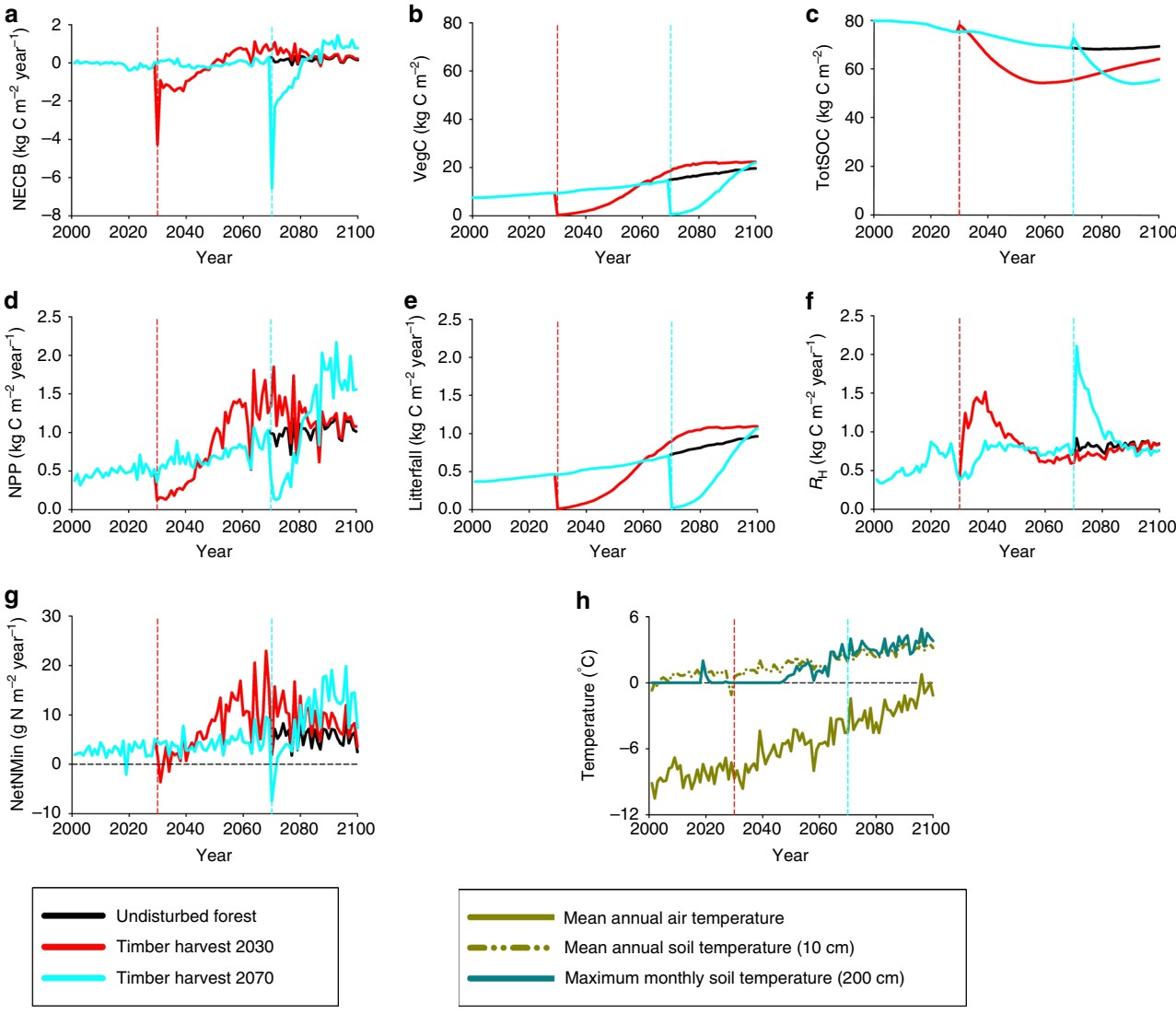

**Fig. 9** Influence of climate change and timber harvest on C and N dynamics of a Siberian larch forest stand during the 21st century under the RCP8.5 scenario. **a** Net C sequestration represented by annual net ecosystem carbon balance (NECB). **b** Vegetation carbon stocks (VegC). **c** Total soil organic carbon (TotSOC) stocks. **d** Annual net primary production (NPP). **e** Annual litterfall carbon. **f** Annual heterotrophic respiration ($R_H$). **g** Annual net N mineralization (NetNMin). **h** Mean annual air temperature, mean annual soil temperature at 10 cm depth, and maximum monthly soil temperature at 200 cm depth. Permafrost thaw is assumed to occur when the maximum monthly soil temperature at 200 cm depth increases above 0 °C. Timber harvests occurred either in the year 2030 (red lines) or 2070 (blue lines). Carbon and nitrogen dynamics of undisturbed forests (no timber harvest) are represented with the black lines

century, but the relative benefits of these enhanced C sinks diminish over time as these secondary forests regrow. As a result, these secondary forests become a less important component of the regional C sink during the latter part of the 21st century.

Other disturbances (wildfires[27,35,36,38,49,50], insect infestations[38], climate-induced vegetation shifts[27], thermokarst ground collapse[51,52]) and mechanisms (SOM decomposition in deep soil[25], weathering of bedrock[53], priming and mineral protection of SOM[54]) not considered in our analyses may be more responsible for any future changes in the overall regional C sequestration in Northern Eurasia. Some of the dynamics found in this study, however, will influence how these other factors affect forest C sequestration.

Consideration of wildfires and insect infestations is particularly important because increases in fires and insect infestations associated with climate change[38,49] could cause Northern Eurasia to become a C source rather than a C sink, even with increased N availability from permafrost degradation and land-use legacy N

subsidies. Previous studies[35,38] have indicated that historical wildfires and insect infestations in Eurasia have led to average annual emissions of about 0.2 Pg C year$^{-1}$ although these emissions can vary by an order of magnitude from year-to-year. Thus, wildfires and insect infestations may lead to a loss of about 20 Pg C over the 21st century to reduce our C sequestration estimates by about one-third (RCP4.5) to one-half (RCP8.5) if there is no change in the frequency, area or severity of wildfires in the future. Unfortunately, the frequency, area and the severity of fires have been noted to be increasing in this region during recent years[5,38]. In a review by Abbott et al.[49], previous studies have also suggested that fire carbon emissions could increase by 200–560% by the end of the 21st century. In this case, our estimates of C sequestration might first appear to be overwhelmed by fire C emissions such that Northern Eurasia would become a C source rather than a sink throughout the 21st century. However, regrowth of forests during recovery after the fire would allow these forests to sequester C, in a manner similar to that described

for a clear-cut timber harvest in Fig. 9. The C sequestered by this forest regrowth would compensate for at least some of the C lost by the fire overall and lead to enhanced C sink activity later during the 21st century[27]. Our analysis would suggest that the relative insensitivity to climate change of forest regrowth on abandoned agricultural land would limit changes in future fire severity in these forests. In contrast, the increase in C sequestered in vegetation in old forests and forests recovering from timber harvest, particularly permafrost forests, in our analysis would suggest that these forests would be vulnerable to more severe fires in the future to enhance C losses. On the other hand, the changes in forest regrowth dynamics from permafrost degradation projected by this study suggests that these forests may recover more rapidly from wildfires in the future because of enhanced N availability and generally more favorable environmental conditions. This benefit would only occur, however, if the frequency and severity of wildfires did not cause an area to become unsuitable for forest growth from green desertification[50], as estimated to have already occurred for tens of millions of hectares in the Russian Far East[5,50].

Climate-induced vegetation shifts may enhance C sequestration with more favorable climate conditions as forests invade areas currently covered by tundra or grasslands, but may enhance C loss in other areas with less favorable climate conditions[27]. Nitrogen subsidies and changing water storage from permafrost degradation may help to speed up the invasion of trees into new areas under favorable climate conditions.

Thermokarst ground collapse occurs as a consequence of permafrost degradation. It substantially enhances the loss of dissolved organic carbon (DOC) and inorganic N such that thermokarst may be a dominant mechanism for transferring C and N from terrestrial ecosystems to river networks with warming[51,52]. While our simulations do account for the general loss of C and N from land ecosystems to neighboring river networks and may account implicitly for some thermokarst effects associated with permafrost degradation, it is not clear how well the overall impacts of this fine-scale disturbance on N availability and C sequestration are represented by our study.

In our analysis, permafrost degradation influences C and N dynamics of SOM in only the top two meters of soil. However, deltaic deposits and Siberian yedoma sediments are estimated to contain a large amount of SOM (about 648 Pg C) at depths greater than three meters[33]. As shown by Koven et al.[25], net N mineralization of this deep soil SOM may provide only a small N subsidy, if any, to vegetation (but see also Hewitt et al.[10]) such that areas covered by these sediments are much more likely to be C sources rather than C sinks, particularly later in the 21st century. Thus, consideration of the decomposition of this deep soil SOM will counteract the enhanced C sequestration activity estimated for these forests by this study such that permafrost forests may not become more important C sinks in the future.

In a recent study, Houlton et al.[53] have indicated that bedrock weathering could also be providing a N subsidy to global taiga forests that is about the same order of magnitude as atmospheric N deposition. While this N subsidy is not considered in our analyses, the exposure of more bedrock to weathering from warmer soil temperatures could enhance the N subsidy to forests from permafrost degradation estimated in our study.

Sulman et al.[54] have indicated that microbe-root interactions, such as priming, may reduce SOM by enhancing decomposition whereas the physical occlusion in microaggregates and chemical sorption in organo-mineral complexes may increase SOM by protecting this SOM from decomposition. They further indicate the relative importance of these compensating effects varies across Northern Eurasia with the gains in SOM being related to decomposition that is limited by cold temperatures rather than

substrate quality. Except for temperature limitation of decomposition, our analysis does not include these effects on SOM so our estimates may be underestimating the loss of C from SOM, the associated increase in N availability, and the size of the N subsidy provided by permafrost degradation.

Incorporation of the disturbance effects and mechanisms described above into model simulations could further improve our understanding of the influence of N availability on land C sink dynamics and improve estimates of C sequestration in the region. In addition, our estimates of forest C sequestration in Northern Eurasia could be improved with a better representation of the effects of past land-use changes and natural disturbances on the current age structure of forests and associated standing stocks of vegetation biomass. Forest inventory data have been useful for providing such improvements in estimates for forests in the United States[28]. Regardless, our study suggests that permafrost degradation and land management decisions will have a large influence on N availability to affect how land C dynamics in Northern Eurasia will evolve in response to future changes in climate, atmospheric chemistry, and disturbances. Thus, carbon–nitrogen interactions need to be considered when assessing sub-regional and regional impacts of global change policies.

## Methods

**Terrestrial ecosystem model**. To explore the relative importance of N subsidies from permafrost degradation, atmospheric N deposition, and land-use legacies on land C sequestration in Northern Eurasia during the 21st century, we conducted a number of simulation experiments using the Terrestrial Ecosystem Model (TEM) driven by climate, atmospheric chemistry and land use data under the RCP 8.5 and 4.5 scenarios[43]. The TEM is a process-based biogeochemistry model that uses spatially referenced information on climate, elevation, soils, and land cover to estimate fluxes and pool sizes of C, N and water in vegetation and soils as influenced by multiple factors such as $CO_2$ fertilization, climate change and variability, land-use change, ozone pollution and atmospheric N deposition[28,36,48,55]. This TEM version (Supplementary Fig. 9) considers the influence on land C dynamics of N inputs from atmospheric N deposition and biological N fixation; losses of N to neighboring river networks in the form of dissolved organic nitrogen (DON) and nitrate; and trace gas losses of N to the atmosphere. In addition to DON, TEM simulates the production and loss of dissolved organic carbon (DOC) to neighboring river networks. This TEM version also simulates the soil thermal regime of an ecosystem[56] so that it is able to simulate the effects of permafrost degradation on land C, N and water dynamics.

**Calculation of land carbon sequestration**. The TEM estimates the net carbon exchange (NCE) of land ecosystems with the atmosphere as follows:

$$NCE = NPP - R_H - E_C - E_L - E_P \qquad (1)$$

where NPP is net primary production (g C m$^{-2}$ mo$^{-1}$), $R_H$ is heterotrophic respiration (g C m$^{-2}$ mo$^{-1}$), $E_C$ is the carbon emissions associated with the conversion of natural land to agricultural land or timber harvests (g C m$^{-2}$ mo$^{-1}$), $E_L$ is the carbon emissions associated with livestock respiration (g C m$^{-2}$ mo$^{-1}$), and $E_P$ is the carbon emissions associated with the decomposition of agricultural and woody products (g C m$^{-2}$ mo$^{-1}$). As described in detail in previous publications[24,36,48], permafrost influences NPP and $R_H$ by affecting soil temperature, the amount of inorganic N available to be taken up by vegetation, the amount of SOM available to decompose, and the amount of liquid water available in soils to support vegetation productivity and SOM decomposition. The permafrost effects on NPP also influence $E_C$, $E_L$, and $E_P$ by affecting the amount of biomass that accumulates in vegetation from NPP over time. Because this version of TEM considers interactions between land and river networks (Supplementary Fig. 9), C sequestration is estimated as net ecosystem carbon balance[57] (NECB, g C m$^{-2}$ mo$^{-1}$) by subtracting the terrestrial loading of DOC to river networks (DOC$_{LOAD}$, g C m$^{-2}$ mo$^{-1}$) from the NCE estimates:

$$NECB = NCE - DOC_{LOAD} = \Delta VegC + \Delta TotSOC + \Delta ProductC \qquad (2)$$

Permafrost also affects DOC$_{LOAD}$ by influencing the amount and timing of runoff[48]. In ecosystems with a shallow active layer, more runoff will occur with snowmelt and less unfrozen DOC will be available to be transported to river networks during the spring than during the summer when the active layer is deeper. In ecosystems with a deeper active layer, more snow melt will remain in the soil to support the decomposition of SOM and the production of DOC, but less water will runoff to limit DOC transport to river networks. In our simulations, DOC is assumed to be refractory C within the soil column and does not decompose. Labile DOC is implicitly included with reactive soil organic carbon (SOC$_R$) and would accordingly decompose within the soil profile. A positive value of NECB

indicates that the land is a C sink (i.e., sequesters carbon) whereas a negative value of NECB indicates that the land is a C source (i.e., loses carbon).

Besides calculating NECB as the sum of ecosystem C fluxes, NECB can also be estimated by summing the changes in land C pools including vegetation carbon ($\Delta$VegC), total soil organic carbon ($\Delta$TotSOC), and C in agricultural and woody products ($\Delta$ProductC) under certain conditions. Equation 2 holds when examining C dynamics for a particular site or for a whole region, but does not hold when assessing the contribution of a particular land cover to regional C source/sink dynamics when the region is undergoing land-use change. This is because the total changes in C stocks aggregated for a particular land cover type in a region includes not only changes in C stocks associated with C sequestration or loss that may occur within that land cover type, but also the changes in C stocks associated with the reassignment of land from one land cover category to another (e.g., conversion of forests to cropland). For example, the changes in total soil organic carbon ($\Delta$TotSOC) for a particular land cover is the sum of changes of new total soil organic carbon occurring within that land cover from C sequestration or loss after the transition between land cover categories ($\Delta$TotSOC$_{NEW}$) and the reassignment of old total soil organic carbon associated with the transition of one land cover to another ($\Delta$TotSOC$_{OLD}$):

$$\Delta\text{TotSOC} = \Delta\text{TotSOC}_{\text{NEW}} + \Delta\text{TotSOC}_{\text{OLD}} \quad (3)$$

Because $\Delta$TotSOC$_{OLD}$ represents the soil organic carbon stock that exists on the land surface both before and after the transition of land from one land cover category to another, $\Delta$TotSOC$_{OLD}$ does not represent a component of land C sequestration, but is rather an accounting tool that describes the gain or loss of C to a land cover type associated with the reassignment of land area that transitioned from or to another land cover type. The use of $\Delta$TotSOC rather than $\Delta$TotSOC$_{NEW}$ to represent C sequestration in soils would overestimate C sequestration in soils in the land cover type that gained area during the land-use change transition and overestimate the C loss from the land cover type that lost area during the transition. While these gains and losses of C balance out at the scale of the entire region, these overestimates of C gains and losses bias the estimated contributions of the underlying land cover types to the regional C source/sink dynamics.

In a similar manner, the changes in vegetation C and agricultural and woody product C pools for a particular land cover type could also be partitioned into a new component from C sequestration and an old component that represents the C reassigned from another land cover type:

$$\Delta\text{VegC} = \Delta\text{VegC}_{\text{NEW}} + \Delta\text{VegC}_{\text{OLD}} \quad (4)$$

$$\Delta\text{ProductC} = \Delta\text{ProductC}_{\text{NEW}} + \Delta\text{ProductC}_{\text{OLD}} \quad (5)$$

Thus, the total amount of carbon sequestered for a particular land cover (NECB) within a region would then be the sum of changes of only the new-carbon components:

$$\text{NECB} = \Delta\text{VegC}_{\text{NEW}} + \Delta\text{SOC}_{\text{NEW}} + \Delta\text{ProductC}_{\text{NEW}} \quad (6)$$

The amount of C sequestered in the vegetation, soil organic matter, and product pools can be estimated from the net balance of the C fluxes into and out of each of these pools for a specified time period:

$$\Delta\text{VegC}_{\text{NEW}} = \text{NPP} - E_{\text{C}} - \text{LtrfalC} - \text{SlashC}$$
$$- \text{NewProductC} - \text{StubbleC} - \text{ForageC} \quad (7)$$

$$\Delta\text{TotSOC}_{\text{NEW}} = \text{LtrfalC} + \text{SlashC} + \text{StubbleC} + \text{ManureC} - R_{\text{H}} - \text{DOC}_{\text{LOAD}} \quad (8)$$

$$\Delta\text{ProductC}_{\text{NEW}} = \text{NewProductC} - E_{\text{P}} \quad (9)$$

where LtrfalC is litterfall carbon; SlashC is the amount of carbon in slash transferred to soil organic matter during the conversion of natural land to agricultural land or timber harvests; NewProductC is the amount of carbon transferred to products from vegetation from the conversion of natural land to agricultural land, agricultural harvest, or timber harvest; StubbleC is the transfer of carbon from crop residues to soils after harvest; ForageC is the transfer of carbon from vegetation to livestock associated with grazing; and ManureC is the transfer of carbon from livestock to soils associated with manure. In our simulations, DOC is assumed to be nonreactive within the soil column and does not decompose. All of these carbon fluxes (g C m$^{-2}$ mo$^{-1}$) have been described in detail in previous publications[26,36,46,48,55,58–65]. Because ForageC is the sum of ManureC and $E_{\text{L}}$, the sum of Eqs. 7, 8, and 9 equals to the sum of the C fluxes described in Eqs. 1 and 2. In our analyses, we estimate C sequestration in vegetation as $\Delta$VegC$_{NEW}$ and C sequestration into SOM as $\Delta$TotSOC$_{NEW}$. Thus, VegC$_{NEW}$ and TotSOC$_{NEW}$ represent the cumulative sum of C sequestered in vegetation and SOM, respectively, over a specified time period. The reassignment of old-carbon in SOM with the transition between land cover types ($\Delta$TotSOC$_{OLD}$) is estimated as the difference between $\Delta$TotSOC and $\Delta$TotSOC$_{NEW}$. Similarly, the reassignment of old-nitrogen in SOM with the transition between land cover types ($\Delta$TotSON$_{OLD}$) is estimated as the difference between $\Delta$TotSON and $\Delta$TotSON$_{NEW}$ where $\Delta$TotSON$_{NEW}$ is estimated as:

$$\Delta\text{TotSON}_{\text{NEW}} = \text{LtrfalN} + \text{AsymNfix} + \text{SlashN} + \text{StubbleN}$$
$$+ \text{ManureN} - \text{NetNMin} - \text{DON}_{\text{LOAD}} \quad (10)$$

where LtrfalN is litterfall nitrogen; AsymNfix is asymbiotic nitrogen fixation (see calculation of nitrogen availability below); SlashN is the amount of nitrogen in slash transferred to soil organic matter during the conversion of natural land to agricultural land or timber harvests; StubbleN is the transfer of nitrogen from crop residues to soils after harvest; ManureN is the transfer of nitrogen from livestock to soils associated with manure; NetNMin is net nitrogen mineralization (see calculation of nitrogen availability below); and DON$_{LOAD}$ is the terrestrial loading of dissolved organic nitrogen (DON) to river networks. All of these N fluxes are in units of g N m$^{-2}$ mo$^{-1}$.

**Calculation of nitrogen availability.** Nitrogen availability in this study is estimated as the sum of net N mineralization, biological N fixation, atmospheric N deposition, and N fertilizer applications. Nitrogen inputs from net N mineralization, biological N fixation, N fertilizer applications are simulated by TEM, but N inputs from atmospheric N deposition of ammonium (NH4dep) and nitrate (NO3dep) are prescribed using spatially explicit time-series data sets. These time-series data sets have been developed by linearly interpolating through time among the gridded (0.5º latitude x 0.5º longitude) snapshots (years 1850, 1980, 2000, 2030, 2100) of the multi-model mean annual NH4dep and NO3dep estimated by Lamarque et al.[66] for the RCP8.5 and RCP4.5 global change scenarios as part of the Atmospheric Chemistry and Climate Model Intercomparison Project (ACCMIP).

Net N mineralization (NetNMin, g N m$^{-2}$ mo$^{-1}$) is estimated as:

$$\text{NetNMin} = \left[ \frac{\text{decay}N_{\text{imm}}\text{VSM}^3[\text{NH}_4]}{k_{\text{n2}} + \text{VSM}^3[\text{NH}_4]} + \frac{\text{SON}_{\text{R}}}{\text{SOC}_{\text{R}}} \right] R_{\text{H}} \quad (11)$$

where decay is the mean decay state of detritus[59] based on the proportion of fast decomposing organic matter to slow decomposing organic matter in the soil and is assumed to be 0.475 (no units) for all ecosystems, $N_{\text{imm}}$ is the maximum rate of soil nitrogen uptake by microbes (i.e., immobilization, g N m$^{-2}$ mo$^{-1}$) that varies by biome and soil texture (Supplementary Table 50), VSM is volumetric soil moisture, [NH$_4$] is the concentration of ammonium nitrogen (NH$_4$-N) available for uptake in the soil solution, $k_{\text{n2}}$ is the Michaelis-Menten half saturation coefficient that is assumed to be 0.0042 g N L$^{-1}$ for all ecosystems, SON$_{\text{R}}$ is the amount of reactive soil organic nitrogen available (g N m$^{-2}$), and SOC$_{\text{R}}$ is the amount of reactive soil organic carbon available (g C m$^{-2}$). Permafrost affects NetNMin by its influence on the amount of unfrozen SON$_{\text{R}}$ and SOC$_{\text{R}}$, along with the amount of liquid water and ammonium in the soil that is available to microbes and by its effects on soil temperature and moisture to influence $R_{\text{H}}$.

Biological N fixation (BiolNfix, g N m$^{-2}$ mo$^{-1}$) is simulated as a linear function of evapotranspiration (ET) using the algorithms of Cleveland et al.[67] adapted to a monthly resolution:

$$\text{BiolNfix} = 0.00102(\text{ET}) + 0.004366667 \quad (12)$$

Nitrogen is added either directly to the vegetation structural nitrogen pool (VegN$_{\text{S}}$) as symbiotic nitrogen fixation (SymNfix) or added to SON$_{\text{R}}$ as asymbiotic nitrogen fixation (AsymNfix) based on the ecosystem partitioning described by Cleveland et al.[67]. The relationship of biological N fixation to evapotranspiration is assumed to be the same for all ecosystems, but the proportion of biological N fixation attributed to symbiotic and asymbiotic N fixation is assumed to vary by biome (Supplementary Table 50). In our simulations, permafrost affects biological N fixation by its influence on soil water supply to support evapotranspiration.

N fertilizer applications (AgFertN, g N m$^{-2}$ mo$^{-1}$) are simulated based on the assumption that N fertilizer is added to croplands such that crop production is never limited by N availability[62]. The amount of added fertilizer is determined by simulating crop production with and without N limitation concurrently and then estimating the optimum amount of N that would need to be added from differences in vegetation uptake of nitrate. No fertilizers are assumed to be applied to croplands before the year 1950. Between 1950 and 1990, the level of fertilizer applications are ramped up by increasing fertilizer levels to be an additional 2.5% of optimum fertilizer levels each year such that an optimum amount of fertilizer is applied to croplands by 1990. After 1990, optimum levels of fertilizers are assumed to be applied to croplands each year and the amount applied varies with changing climate and atmospheric chemistry conditions. Our estimate is compared to that derived from Galloway et al.[6] for the 1990s if fertilizer applications in Europe and the countries comprising the former Soviet Union are assumed to be the difference between fertilizer exports (13.2 Tg N year$^{-1}$) and the sum of fertilizer imports (6.6 Tg N year$^{-1}$) and fertilizer production (21.6 Tg N year$^{-1}$) in the region.

Permafrost affects these optimum N fertilizer applications based on its influence on crop productivity and net N mineralization. Cool soil temperatures hinder crop productivity and thereby restrict the amount of N fertilizers needed to support that reduced productivity. On the other hand, warming enhances crop productivity to increase crop N demands, but this demand might be compensated by concurrent increases in net N mineralization of croplands such that less N fertilizers are required.

**Development of regional estimates**. Monthly land C and N fluxes and pools are estimated by TEM for 17,773 grid cells (0.5° latitude × 0.5° longitude) comprising Northern Eurasia during the time period from year 1500 to 2100. The TEM is driven by projections of climate (surface solar radiation, air temperature, precipitation), atmospheric carbon dioxide and ozone concentrations from simulations developed with the MIT IGSM-CAM model under the RCP8.5 and RCP4.5 scenarios[44] (Fig. 2). Specifically, this study relies on a single simulation for each RCP scenario from a large ensemble with different initial conditions and different values of climate sensitivity. Here, we use simulations that assumed the median value of 3.0 °C for climate sensitivity, but recognize that there is substantial uncertainty in future projections of climate change in the region[44] that we are not exploring in this study. Land-use change projections are based on Hurtt et al.[68] land-use transitions. Because most of the land use in the RCP8.5 scenario is timber harvest, we tracked the C and N dynamics of young forests (stand ages less than 120-years-old) separately and in combination with old forests (stand ages either equal to or greater than 120-years-old) based on the analyses of the influence of stand age on C sequestration by Pregitzer and Euskirchen[69].

**Representation of permafrost degradation**. To examine how permafrost degradation may influence N availability and C sequestration in Northern Eurasian ecosystems, we define permafrost ecosystems to be those land covers in which the simulated maximum monthly soil temperature at 200 cm never got above 0 °C during the year 2000. With this definition, permafrost ecosystems are estimated to cover about 920 million ha. The changes in land use over the 21st century for these permafrost ecosystems is presented in Supplementary Table 45. The corresponding ecosystem responses of N availability and C sequestration to the two global change scenarios projected by TEM for these ecosystems are presented in Supplementary Tables 46–49. To examine how permafrost degradation may have influenced the extent of permafrost over the 21st century, we also determine the area of the land covers in which the simulated maximum monthly soil temperature at 200 cm never got above 0 °C during the years 2050 and 2100. The extent of permafrost in Northern Eurasia is estimated to decline to 664 million ha under the RCP4.5 scenario and 326 million ha under the RCP8.5 scenario by the end of the 21st century (Supplementary Fig. 1, Supplementary Table 4). Most of this change in permafrost extent occurs after 2050 under the RCP8.5 scenario, but before 2050 under the RCP4.5 scenario.

**Representation of land-use history**. To explore how N subsidies from land-use legacies may influence forest C sequestration and N availability, we examine the influence of both cropland abandonment and pasture abandonment on forest C and N dynamics. A dynamic cohort approach[70] has been adopted to represent land-use history in TEM. In this approach, TEM assumes that the entire area of each of the 17,773 grid cells in Northern Eurasia is initially covered by a mosaic of undisturbed potential vegetation cohorts. When a disturbance occurs, such as timber harvest or the conversion of land to croplands or pastures, a new cohort is formed from one of the undisturbed potential vegetation cohorts and the disturbed land area within the grid cell is then subtracted from that undisturbed potential vegetation cohort and assigned to the new disturbed cohort. As time progresses in the TEM simulation and more disturbances occur, more cohorts are added to the grid cell from either undisturbed potential vegetation or previously disturbed cohorts if the disturbance does not affect the entire area covered by the previously disturbed cohort. Otherwise, the land cover of the existing disturbed cohort is just reassigned to the new land cover. Cohorts are also added when croplands or pastures are abandoned if the area abandoned does not affect the entire area of the cropland or pasture cohort. As each disturbance is tracked separately within TEM, different types of disturbances within a grid cell can be considered simultaneously and allows TEM to consider the impacts of multiple disturbances on land C and N dynamics. The timing, location and affected area of a disturbance are prescribed by a spatially explicit time-series land cover data set developed from the land transitions described by Hurtt et al.[68] from the year 1500 to the year 2100.

Within TEM, disturbance-related C and N fluxes are calculated and land C and N stocks are adjusted within the newly disturbed cohort to account for the initial effects of the disturbance prescribed by the land cover data set. In later years, the TEM is then used to simulate the C and N dynamics of the disturbed cohort and the recovery of land C and N dynamics after abandonment within the context of local environmental conditions.

**Importance of nitrogen subsidies on forest carbon sinks**. To quantify and evaluate the relative importance of N subsidies from atmospheric N deposition, permafrost degradation, and land-use legacies (including the effects of fertilizer applications to croplands) on regional C sequestration in forests, several simulation experiments have been conducted. In addition, a site-level simulation experiment is conducted to better understand how warming and permafrost degradation may interact with the recovery of permafrost forests after timber harvests in Siberia under the RCP8.5 scenario.

**Atmospheric nitrogen deposition**. To evaluate the importance of N subsidies from atmospheric N deposition on forest C sequestration and N availability, we repeated the simulations using the RCP8.5 scenario and RCP4.5 scenario, but

replaced the atmospheric N deposition data sets from these scenarios with a data set where annual atmospheric N deposition remained constant at the levels estimated to occur during the year 1850 in each grid cell. The atmospheric N deposition effects on various C and N fluxes are determined by subtracting the TEM estimates determined using the constant atmospheric N deposition data set (Supplementary Tables 18, 24, 32 and 37) from the corresponding TEM estimates determined when atmospheric N deposition is allowed to change over time (Supplementary Tables 5, 6, 7, and 9).

**Permafrost degradation**. To evaluate the importance of N subsidies from permafrost degradation on forest C sequestration and N availability, we modified TEM such that the seasonal active layer depth varied during the historical portion of the simulation, but then remained constant after the year 2000. In TEM, the active layer depth determines the proportion of soil organic C and N that is exposed to decomposition/mineralization[24]. In a normal TEM simulation, the active layer is allowed to vary seasonally and allowed to deepen in response to warming soil temperatures. Permafrost degradation effects on various C and N fluxes are determined by subtracting the TEM estimates determined using the constant seasonal active layer depth (Supplementary Tables 13, 19, 27, and 33) from the corresponding TEM estimates determined when the active layer depth is allowed to deepen (Supplementary Tables 5, 6, 7, and 9).

**Land-use legacies including fertilizer applications**. To evaluate the importance of N subsidies from land-use legacies, we consider N subsidies associated with the abandonment of both croplands and pastures to forests. For legacies of cropland abandonment in this study, we identified cohorts within each of the 17,773 grid cells that are covered by croplands during the year 2000, if they existed, and then follow the fate of those cohorts throughout the 21st century (Supplementary Table 25) as described in the Data analysis section below. The annual fluxes of young forests are then summed to determine cumulative estimates of NetNMin, NECB, VegC$_{NEW}$, TotSOLC$_{NEW}$, NPP, and $R_H$ over specified time periods for secondary forests growing on abandoned croplands. We also examine how fertilizer applications may have modified the response of forests to cropland abandonment. As described in the calculation of nitrogen availability section above, we assume that fertilizers are applied to croplands after 1950 with optimal levels applied after 1990 in our normal TEM simulations.

To evaluate the importance of N subsidies from legacy N fertilizer applications on forest C sequestration and N availability, we conducted two additional simulations each for the RCP8.5 and RCP4.5 scenarios. In the first simulation, we modified TEM such that no fertilizers are ever assumed to be applied to croplands. In the second simulation, we modified TEM such that fertilizers are applied until the year 2000, but then no fertilizers to croplands are applied after the year 2000. To estimate the effect of future fertilizer applications (i.e., a land management approach that could be modified in the future) on forest C and N fluxes, we subtract the TEM estimates determined using the pre-2001 fertilizer applications (Supplementary Tables 15, 16, 21, 22, 29, 30, 35, 36, 40, 41, 43, and 44) from the corresponding estimates of our normal TEM simulations (Supplementary Tables 14, 20, 28, 34, 39, and 42). To estimate the effect of past fertilizer applications (i.e., a legacy of past land management that cannot be modified in the future), we subtract the TEM estimates determined from the unfertilized cropland simulation from the corresponding TEM estimates from the simulation using pre-2001 fertilizer applications for forests regrowing on land that was cropland during the year 2000 (Supplementary Tables 15, 16, 21, 22, 29, 30, 35, 36, 40, 41, 43, and 44).

For legacies of pasture abandonment in this study, we identified cohorts within each of the 17,773 grid cells that were covered by pastures during the year 2000, if they existed, and then followed the fate of those cohorts throughout the 21st century (Supplementary Table 26) as described in the Data analysis section below. The annual fluxes of young forests are then summed to determine cumulative estimates of NetNMin, NECB, VegC$_{NEW}$, and TotSOLC$_{NEW}$ over specified time periods for secondary forests growing on abandoned pastures. Because pastures are never assumed to be fertilized, our estimates of C and N fluxes in these secondary forests (Supplementary Tables 17, 23, 31, and 38) are based on our normal TEM simulations.

**Effect of warming and timber harvest on permafrost forests**. To examine why permafrost forests appear to have a very different response to timber harvests during the second half of the 21st century than during the first half of the century, a simulation experiment has been conducted based on harvesting a larch forest stand underlain by permafrost during the year 2000. In the experiment, we use the simulated climate and atmospheric chemistry conditions under the RCP8.5 scenario for a grid cell at 65.0° N latitude and 125.0° E longitude with three timber harvest scenarios (no timber harvest, a timber harvest during the year 2030, a timber harvest during the year 2070). The site is assumed to be covered by undisturbed natural vegetation until the timber harvest occurs. The site is also underlain by permafrost until the year 2047 when maximum soil temperatures at 2 meters depth begin to steadily increase above 0 °C (Fig. 9h).

**Data analysis**. To analyze TEM output, a number of customized programs have been developed to process the gridded model estimates into regional summary estimates and maps. First, a customized C++program (xtranalchrt606b) is used to determine regional estimates stratified by land-cover type and year. This program can determine a time series of the total area covered by each land-cover type in the region each year along with the corresponding aggregated sum of C or N fluxes, or C or N pool sizes, for that land-cover type for a specified month or annual estimates based on the TEM variable represented in the output file. The program also allows these summary estimates to be developed for a specified range of stand ages based on the cohort information within each of the grid cells. For this study, xtranalchrt606b is used to develop summary estimates of annual C and N fluxes, and December C and N pool sizes of each year for all land cover types and for young (less than 120-year-old) forest types. The time-series summary estimates from this program are then re-organized into Excel files to further aggregate the fluxes and pool sizes into broader land cover categories (e.g., forests from temperate broadleaved deciduous forests, temperate evergreen needleleaf forests, boreal evergreen needleleaf forests, boreal deciduous needleleaf forests, etc.) and to quantify cumulative C and N fluxes, and changes in C and N pool sizes over specified time periods. This reorganization occurs using the program createIGSMTEMfluxExcelSummary.py for flux variables (e.g., NPP) estimated by TEM and createIGSMTEMpoolExcelSummary.py for pool variables (e.g., VegC) estimated by TEM. To evaluate land-use history, changes in area covered by the broader land cover categories over time are estimated using the program sumIGSMVEGregion.py. The resulting aggregated time-series data from these programs customized for TEM are then imported into SigmaPlot 12.5 to produce graphics and determine statistical analyses,

For climate and atmospheric chemistry input data, Python programs have also been developed and used to determine a time-series of mean annual estimates (e.g., sumAveClmRegion.py for air temperature, incoming solar radiation, AOT40) or total annual sum estimates (e.g., sumAnnPrecRegion.py for precipitation) for the entire region from the gridded data. Although atmospheric N deposition is an input to TEM, it has also been set up as a TEM output so that the N input into different land covers from this N source could be evaluated using xtranalchrt606b.

Besides summary estimates, Python programs have been developed and used to create ASCII files that map the spatial distribution of cumulative C and N fluxes from all ecosystems over a specified time period (mapCumFluxRegion.py), C and N pool sizes for a particular year (mapPoolRegion.py). In addition, Python programs have been developed and used to focus on mapping C and N fluxes from forests (mapForestCumFluxRegion.py, mapForestPoolRegion.py). Besides C and N fluxes and pool sizes, programs have been developed and used to map the distribution of mean soil temperatures at specific depths and specified time periods (mapDSTregion.py) and the maximum soil temperatures for a particular year (mapPermafrostRegion.py) from the gridded TEM output files. As described earlier, permafrost is assumed to occur where the maximum monthly soil temperature at 200 cm depth never gets above 0 °C during the specified year. A C++program (mapVegCoverage606c) has also been developed and used to map the proportional coverage in a grid cell of specified land covers for a specified range of stand ages during a particular year from the land cover data sets. The ASCII map files resulting from these programs are able to be read directly into ArcMap 10.6 to create the maps displayed in this study. Another C++ program (diffmap606c) has been developed and used to map changes in the distribution of land cover or changes in C and N pool sizes over the years by calculating the difference between the underlying maps described by the ASCII files. Similarly, a C++program (addmap606c) has been developed and used to map the addition of estimates in two underlying maps described by the ASCII files, such as the distribution of biological N fixation from the sum of the distribution of symbiotic N fixation and the distribution of asymbiotic N fixation. Finally, customized programs have been developed and used to map cropland abandonment (mapCrop2ForestRegion.py) based on changes in the distribution of both croplands (losses) and young forests (gains) and to map pasture abandonment (mapPasture2ForestRegion.py) based on changes in the distribution of both pastures (losses) and young forests (gains). After mapping cropland and pasture abandonment, the distribution of timber harvest is estimated by using diffmap606c to subtract the proportion coverage of young forest gained from cropland and pasture abandonment from the overall changes in coverage by young forests over the specified time period. Negative changes in the proportion coverage of young forests over a time period are assumed to be caused by forest aging to become old forests.

To examine how permafrost degradation affects C and N dynamics in ecosystems underlain by permafrost, a customized program (createPermafrostMask.py) has been developed and used to create a list of locations (longitude, latitude) of grid cells where permafrost is assumed to occur. This list is used by a second program (getDataSubset.py) to obtain a subset of TEM output where permafrost is assumed to occur during the years 2000, 2050, and 2100.

To examine the influence of the abandonment of agricultural land on terrestrial biogeochemistry during the 21st century, a couple of Python programs (abandonedcrop2000regionTEMflux.py, abandonedcrop2000regionTEMpool.py) have been developed and used to identify the cohorts that were croplands during year 2000 and then track the C and N dynamics of these particular cohorts throughout the 21st century regardless if these cohorts remain croplands, are converted to pastures, or are abandoned to natural vegetation. The output from these programs contain summary estimates that describe the fate of these cohorts

to future land covers. These estimates include annual changes in both the areas of various land cover categories (i.e., food crops, biofuel crops, pastures, young forests, old forests, all forests, grasslands, shrublands, tundra, wetlands, deserts) covered by these cohorts and either the associated total annual flux or December pool sizes. Note that no croplands are assumed to be explicitly producing biofuels in this study. A similar set of Python programs (abandonedpasture2000regionTEMflux.py, abandonedpasture2000regionTEMpool.py) have been developed that identify the cohorts that were pastures during year 2000 and then tracked the fate of these cohorts throughout the 21st century with the same output of summary estimates as abandoned croplands.

Finally, to examine how land management decisions of the two global change scenarios may influence forest structure in the future to affect terrestrial biogeochemistry, a Python program (standAgeDistribution.py) has been developed and used to estimate the distribution of stand ages of forests in a region for a specified year based on the cohort information contained within each grid cell. In the output, stand ages are aggregated in 10-year increments from a stand age equal to zero to a stand age equal to 350-years-old. All forests older than 350-years-old are then aggregated together in the oldest stand age category.

## Data availability
The underlying data supporting the findings of the study are available from the corresponding author upon reasonable request.

## Code availability
The TEM C++ source code, including code modifications to examine permafrost degradation effects and land-use legacy effects in this study, is publicly available from [https://github.com/MBL-TEM/TEM6.0.6] for educational and research purposes. In addition, the customized programs used to process TEM output are also available in either the src or the Utilities sub-directories of the GitHub repository.

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

## Acknowledgements

This research was supported by the US National Aeronautics and Space Administration (NASA) Land-Cover and Land-Use Change (LCLUC) Program grant NNX14AD91G. The Joint Program on the Science and Policy of Global Change is funded by a number of

federal agencies and a consortium of 40 industrial and foundation sponsor (for the complete list see http://globalchange.mit.edu/sponsors).

## Author contributions

D.W.K. and J.M.M. conceived the paper, analyzed the data, and wrote the main paper and Supplementary Information. Q.Z. wrote the TEM code describing soil thermal dynamics and their effects on land carbon and nitrogen dynamics. D.W.K. wrote part of the TEM code associated with N inputs and losses and ran the TEM model for the analysis. E.M. and A.P.S. wrote the code and ran the IGSM-CAM model to create the input data for the TEM simulations. D.W.K. also developed the code used to process the TEM results for the analysis. All authors discussed the results and implications and commented on the manuscript at all stages.

## Additional information

**Competing interests:** The authors declare no competing interests.

