## [Peer Review File · Nature Communications]

Reviewers' comments:

Reviewer #1 (Remarks to the Author):

General comments:

This manuscript describes a modeling study of carbon sequestration in Northern Eurasia. The unique angle of this study is that it links ecosystem nitrogen dynamics to carbon sequestration in global/regional models of carbon balance. This is of paramount importance for both basic research in global change biology and provides new estimates of carbon sequestration that are consistent with our current understanding of coupled biogeochemical cycles. These estimates should also be applicable to reviews for policy makers.

The manuscript is clearly written and presents exciting results in the body of the text and supplement. However, I found that the important contributions of this manuscript failed to emerge as clearly as I would have liked; the results and conclusions reported in the introductory paragraph were vague. Despite the topic sentence, the authors do not attribute the nitrogen effect on carbon sequestration in a quantitative way. Up front, I would like to know how much of the predicted future carbon sink is attributable to (1) enhanced N mineralization from cold soils, and (2) the legacy of past nitrogen fertilizer use. For (1), I found the arguments for permafrost N tenuous and not well developed in the main manuscript; it was more detailed in the supplement. If this is one of the first order conclusions, then I would like to see the rationale more developed in the main text. For (2), the important results are buried in the text; 45% of all forest carbon sequestration attributable to the legacy of past fertilization! This is exciting, but the fact that it is not included in the introduction gives me pause. Were the authors cautious or uncertain about the relative magnitudes of these effects? Finally, there is considerable text allocation to findings about forest harvest, yet this is not included in the introduction.

In the concluding paragraph, the authors list a number of important caveats. One of the most important is the exclusion of fire in a region where it has a large impact on forest structure and ecosystem processes. I would like to see a more developed rationale for why these results are important given that the very large influence of fire is not included. Given the explicit treatment—and important effects—of forest harvest, might it be possible to extend inference to fire? Alternatively, would it be possible to discuss how the primary findings of the paper might be altered (or not) by the inclusion of fire? For example, the effects of legacy fertilizer on carbon sequestration might not be impacted by fire because of the young age or the mesic precipitation status of these forests. Without more explicit discussion of fire, it is difficult for me to see the generality and importance of the regional estimates.

Overall, I was left questioning the important results from this paper. I would like to see a clear statement of how these results advance both conceptual and quantitative understanding of Eurasian carbon sequestration in future climates. I would also like to see more developed logic for how results should be interpreted in the absence of fire.

Specific comments:

Introductory paragraph

How does this new estimate of carbon sequestration compare to the 20th century sink? How does this compare to past estimates? How should this inform our understanding?

What are the spatial constraints on Northern Eurasia? What are the biomes? Looks like all temperate and boreal forests and tundra in Eastern Europe, Scandinavia, FSSU, China and Russia.

As a reader, I want to immediately understand how these results affect my understanding of Eurasian forest and tundra carbon sequestration, but I cannot do this without more context. This context is partially embedded in the first sentence of the supplement.

“hot spots” is used in the introductory paragraph but never again. If this is important enough to use in quotations, it should be important enough to include in the manuscript.

Permafrost forests? “In 2000, we estimate that 29% of all forests in Northern Eurasia were underlain by permafrost (Supplementary Fig. S4).” This figure only shows permafrost distribution, not forest distribution. Could you overlay the forest distribution on the permafrost map so that the reader can see the distribution of that 29%?

How is fire represented? This is likely to have large effects on forest age structure under the different climate scenarios. I understand that it is not included as a dynamic process, but do you

change the age structure of the forests under the different scenarios to reflect more frequent disturbances? I see from the caveats at the end of the paper that this is not included.

Fire should have similar impacts to timber harvest in some of the regions. Understory fire versus stand-replacing?

What are the relative impacts of temperature and moisture on forest growth? How does the model represent drought index under the different climate scenarios? Seems like this is key to predicting forest growth. What is the relative important of temperature versus precipitation differ across regions and climate scenarios? How does the 'wetness' of these climate scenarios compare to other modeling efforts?

"Over the 21st century, we project Northern Eurasian ecosystems will sequester 40.6 Pg C under the RCP8.5 scenario and 53.7 Pg C under the RCP4.5 scenario." So this is about 0.4 and 0.5 Pg C yr⁻¹, which is within the range of estimates for current rates. Why is this exciting? Is the excitement in the spatial distribution of the sinks and sources?

"Under the RCP8.5 scenario, increases in carbon sequestration rates (0.0040 ± 0.0004 Pg C yr⁻², $r^2 = 0.47$, $p < 0.0001$, $n = 100$) of the undisturbed old forests are supported by concurrent warming-induced increases in net N mineralization (0.065 ± 0.011 Tg N yr⁻², $r^2 = 0.27$, $p < 0.0001$, $n = 100$) throughout the 21st century." Not clear to me why this correlation is meaningful—could be driven by model structure. Please explain the independence of these estimates and why this might be representative of something other than model structure.

Page 6, Second paragraph.

Changes to carbon sequestration in permafrost forests gets a fair amount of attention in the main body of the manuscript, yet there are no primary figures that provide direct support for changes in these forests. They are buried in the supplementary information.

How important are these values relative to regional estimates?

How novel are these mechanisms or conclusions?

How does this knowledge change our understanding?

How useful is modeling of forest carbon balance that does not include fire, or pests and diseases?
How useful is modeling of permafrost carbon balance that does not include carbon stocks below 2 m depth?

Figure 1 caption: loss should be lost, or alternatively, gained should be gain to match tenses. I find the comparison between the nitrogen circles and the carbon bars difficult to interpret. There is a lot of information contained in this figure, but not much of it seems salient to the text. I would like to see the relative partitioning of regional carbon fluxes in one figure or table rather than having them spread out through 9 figures. I also find that the nitrogen circles are hard to examine. I appreciate the idea that nitrogen cycles, but I would like to see something more quantitative, where I could easily extrapolate data to compare the absolute and relative magnitude of effects. At the very least, I would like to see estimates of uncertainty on these figures.

Figure 2. Estimates of uncertainty? If fire were included in these temporal trends, how might they be different? My guess is that the permafrost forests—largely fire-prone—would not show high forest sequestration.

Figure 4. If fire—both understory and stand-replacing—were included in this figure, how might it be different?

Reviewer #2 (Remarks to the Author):

The manuscript by Kicklighter et al shows projections of carbon balance for the northern Eurasian region over the 21st century using the TEM ecosystem model. The model has been well-tested in high latitude ecosystems, with an interesting representation of how permafrost, nitrogen, and land management dynamics all interact with the carbon cycle. The interesting result here is that nitrogen, land management, and recovery from past disturbances play a relatively large part in the carbon dynamics of the region. Typically people tend to focus on the carbon stored in permafrost soils as the dominant driver of carbon change in the region, but the simulations suggest that, at least over the period of interest here, nitrogen plays a key role and the land use dynamics are also crucial.

My main concern here is on the interpretation of the results: how do the authors know that it is nitrogen from warming soils, rather than direct effects of warming on longer growing seasons, that drives the carbon uptake? The same argument applies to land-use: how do they point to past fertilizer use on abandoned lands, rather than the land abandonment itself, as the driver of carbon uptake? The figures are making a correlative case for this, but the great strength of models like this is that you can unambiguously attribute the drivers of a given change by comparing the response of the system to various forcing scenarios. So to really argue for the importance of nitrogen, it seems like you would need to do an experiment where you hold nitrogen mineralization fixed and allow everything else to change; with that you could identify the role of nitrogen within the suite of other drivers. I don't see any attempt made to do this kind of experiment. Thus the conclusions are somewhat speculative.

Otherwise the paper is interesting and I see it as a worthwhile contribution to the literature of carbon cycle responses to global change at high latitudes.

One final issue is that the data availability statement is weak: it is (like it or not) 2018 and I think the expectation should be that one archives one's results publicly at the time of publication, rather than simply stating that they are available upon reasonable request.

Reviewer #3 (Remarks to the Author):

Summary: Kicklighter et al. use the TEM model of coupled terrestrial C-N cycling to predict rates of terrestrial C storage across northern Eurasia of future two contrasting scenarios (RCP 4.5 and 8.5) for changes in climate, atmospheric chemistry (CO₂, N deposition) and land use over the next century. They predict that warming-driven increase in soil N mineralization will stimulate net carbon uptake by the region's forests, with additional growth driven by assumed fertilization of croplands later abandoned.

General comments: The paper is clear and easy to read. These simulations appear to be fairly straightforward model projections of regional (N. Eurasia) carbon balance in response to IPCC-projected scenarios, using an established biogeochemistry model. As presented, this model seems structured to yield results generally similar to most other terrestrial ecosystem and earth system models applied to examine terrestrial C storage in response to this suite of projected environmental changes.

For example, TEM and other models structured with classic first-order soil decomposition dynamics generally predict that warming will increase soil N availability to plants that will offset the C losses from soils (e.g., Bonan 2008 *Nature Geoscience*); it's not wholly apparent if or how this model application yields results that are especially unusual or exceptionally large, or marks a substantial advance in modeling approaches. Other high-profile advances in soil C-N modeling are exploring the role of microbial dynamics with priming or mycorrhizal allocations to enhance plant acquisition of soil N (e.g., Sulman et al. 2014, *Nature Climate Change*, Wieder et al. 2018 *Global Change Biology*), a set of processes that are likely to be very important in supplying N to Eurasia's boreal forests, but not yet included in this model analysis.

If this analysis aimed to highlight inclusion of its permafrost model as a novel module, that contrast over past work by this or other models should receive greater emphasis – e.g., contrasting results with and without that module, and comparisons with observations to confirm model performance or identify areas of continued challenge. The role of N from fertilization of cropland later abandoned to forest is interesting and potentially novel, but seems to be largely assumed input of N fertilizer rather than values supported by fertilization data; additional support for that assumption would bolster the model's corresponding conclusions about the role of that N.

Overall, this manuscript would also generally benefit by the addition of some comparisons with observations of important C and/or N cycle pools or fluxes, both to demonstrate model performance under current or altered (experimental) conditions and to yield other insights for model improvement in the future.

Response to Reviewers' Comments

Reviewers' comments:

Reviewer #1 (Remarks to the Author):

General comments:

This manuscript describes a modeling study of carbon sequestration in Northern Eurasia. The unique angle of this study is that it links ecosystem nitrogen dynamics to carbon sequestration in global/regional models of carbon balance. This is of paramount importance for both basic research in global change biology and provides new estimates of carbon sequestration that are consistent with our current understanding of coupled biogeochemical cycles. These estimates should also be applicable to reviews for policy makers.

The manuscript is clearly written and presents exciting results in the body of the text and supplement. However, I found that the important contributions of this manuscript failed to emerge as clearly as I would have liked; the results and conclusions reported in the introductory paragraph were vague. Despite the topic sentence, the authors do not attribute the nitrogen effect on carbon sequestration in a quantitative way. Up front, I would like to know how much of the predicted future carbon sink is attributable to (1) enhanced N mineralization from cold soils, and (2) the legacy of past nitrogen fertilizer use. For (1), I found the arguments for permafrost N tenuous and not well developed in the main manuscript; it was more detailed in the supplement. If this is one of the first order conclusions, then I would like to see the rationale more developed in the main text. For (2), the important results are buried in the text; 45% of all forest carbon sequestration attributable to the legacy of past fertilization! This is exciting, but the fact that it is not included in the introduction gives me pause. Were the authors cautious or uncertain about the relative magnitudes of these effects? Finally, there is considerable text allocation to findings about forest harvest, yet this is not included in the introduction.

In the concluding paragraph, the authors list a number of important caveats. One of the most important is the exclusion of fire in a region where it has a large impact on forest structure and ecosystem processes. I would like to see a more developed rationale for why these results are important given that the very large influence of fire is not included. Given the explicit treatment—and important effects—of forest harvest, might it be possible to extend inference to fire? Alternatively, would it be possible to discuss how the primary findings of the paper might be altered (or not) by the inclusion of fire? For example, the effects of legacy fertilizer on carbon sequestration might not be impacted by fire because of the young age or the mesic precipitation status of these forests. Without more explicit discussion of fire, it is difficult for me to see the generality and importance of the regional estimates.

Overall, I was left questioning the important results from this paper. I would like to see a clear

statement of how these results advance both conceptual and quantitative understanding of Eurasian carbon sequestration in future climates. I would also like to see more developed logic for how results should be interpreted in the absence of fire.

We see four main issues to be addressed in the above comments:

Comment 5: Vague results and conclusions reported in the introductory paragraph (now Abstract).

Response 5: With the reorganization of the manuscript, we substantially revised the Abstract to indicate the important contributions of the study; i.e., the importance of nitrogen subsidies from permafrost degradation, land-use legacies, and atmospheric N deposition on carbon sequestration in the area, which accounts for 33-52% of the net sink in the region. We further indicated that the relative importance of permafrost degradation and land-use legacies differed between the two climate scenarios. We did not further quantify these effects or attempt to describe the interaction of permafrost degradation with timber harvest response in the Abstract due to word limitations.

Comment 6: Authors do not attribute the nitrogen effect on carbon sequestration in a quantitative way.

Response 6: As described in **Response 2**, we conducted a number of additional simulations to quantify the size of nitrogen subsidies provided by permafrost degradation, land-use legacies, and atmospheric N deposition and their effect on forest carbon sequestration. The quantitative results of these simulations and their attribution to the various factors are summarized in the new Table 1. By quantifying the effects of land-use legacies, we found that fertilizer applications to croplands accounted for 10.8% of the carbon sequestration by forest trees regrowing on abandoned croplands, but had relatively little importance in the overall magnitude of the carbon sequestered by all forest vegetation (2.9% under the RCP4.5 scenario, lines 165-169), but influenced the timing of carbon sink activity such that secondary forests growing on abandoned fertilizer croplands were larger carbon sinks later in the century than if the croplands had not been fertilized (lines 204-215). Most of the land-use legacy effects on carbon sequestration by forest under the RCP4.5 scenario is a result of the “unfertilized” component of the nitrogen subsidies associated with the large area of cropland and pastures abandoned under this scenario (lines 162-165 and Table 1).

Comment 7: How might the primary findings of the paper be altered by the inclusion of fire?

Response 7: In lines 388-417 of the Discussion, we examine how consideration of wildfire effects may alter our estimates of regional carbon sequestration with the region becoming more of a carbon source and less of a sink. In addition, we also note that insights gained from our analyses suggest that the severity of future fires will likely increase in the future to increase the loss of carbon from wildfires in most forests, particularly in permafrost forests. However, we note that subsequent recovery of these forests from wildfires may occur more rapidly to increase carbon sink activity later in the century as a result of more favorable climate conditions and nitrogen subsidies from permafrost degradation – similar to the ecosystem responses to timber harvest. In contrast, we note that fire severity may not increase in secondary forests growing on abandoned agricultural land as carbon sequestration in these forests appear to be much less sensitive to climate change.

Comment 8: Provide a more developed rationale for the study and the importance of its findings (forest harvest, warming-enhanced N mineralization, legacy of past nitrogen fertilizer use) in advancing conceptual understanding of Eurasian carbon sequestration.

Response 8: As indicated in **Response 1**, we have reorganized the manuscript to focus the manuscript on the effects of permafrost degradation, land-use legacies, and atmospheric N deposition on nitrogen availability to influence carbon sequestration in the region. We modified the text (lines 41-49) to indicate that the effects of permafrost degradation and land-use legacies have not been examined in previous studies (although there have been hints of such effects) and that we were going to assess the consequences of these factors by quantifying the effects in this study. In Figures 3 and 4, we show that carbon sequestration in the region is dominated by forests and that nitrogen availability is dominated by net nitrogen mineralization. Thus, as described in lines 114-116, we focus our analyses on the influence of permafrost degradation and land-use legacies on net nitrogen mineralization and carbon sequestration in forests. We then describe how permafrost degradation and land-use legacies provide nitrogen subsidies to influence carbon sequestration in forests and how they may compare to nitrogen subsidies from atmospheric N deposition in lines 119-133.

In the Discussion, we summarize the importance of the study's findings in lines 359-381 before acknowledging some of the limitations of the modelling approach (lines 381-385) and then discuss how consideration of other factors influencing carbon sequestration in Northern Eurasia may modify the estimates of carbon sequestration determined in this study and also how the some of the dynamics discovered in this study may influence the response of Northern Eurasian ecosystems to those other factors (lines 385-453).

Specific comments:

Introductory paragraph

Comment 9: How does this new estimate of carbon sequestration compare to the 20th century sink? How does this compare to past estimates? How should this inform our understanding?

Response 9: The Introductory paragraph (now the Abstract) has been substantially modified to highlight the relative importance of nitrogen subsidies from permafrost degradation, land-use legacies, and atmospheric N deposition on carbon sequestration in Northern Eurasia. As described in **Response 1**, these account for 33% [RCP8.5] to 52% [RCP4.5] of the net carbon sink with nitrogen subsidies from permafrost degradation being more important under RCP8.5 scenario and nitrogen subsidies from land-use legacies being more important under the RCP4.5 scenario.

Comment 10: What are the spatial constraints on Northern Eurasia? What are the biomes? Looks like all temperate and boreal forests and tundra in Eastern Europe, Scandinavia, FSSU, China and Russia.

Response 10: We added a new Figure 1 to the manuscript to identify the spatial extent of Northern Eurasia plus a pie chart to show the relative area of the different biomes within the region. Maps of the spatial distribution of these biomes are still provided in Supplementary Figure S1. A list of countries within Northern Eurasia with forest carbon stocks (except China) as estimated by FAO is given in a new Supplementary Table S2b.

Comment 11: As a reader, I want to immediately understand how these results affect my understanding of Eurasian forest and tundra carbon sequestration, but I cannot do this without more context. This context is partially embedded in the first sentence of the supplement.

Response 11: As indicated in **Response 1**, the manuscript has been reorganized to emphasize the relative importance of nitrogen subsidies from permafrost degradation, land-use legacies, and atmospheric nitrogen deposition on forest carbon sequestration, which dominates carbon sink dynamics in Northern Eurasia. As a result, the Abstract (formerly Introductory paragraph) has been substantially changed to reflect this change in context and should now indicate how the study results might affect understanding of carbon sequestration in Northern Eurasia.

Comment 12: “hot spots” is used in the introductory paragraph but never again. If this is important enough to use in quotations, it should be important enough to include in the manuscript.

Response 12: We deleted the term “hot spots” from the revised manuscript.

Comment 13: Permafrost forests? “In 2000, we estimate that 29% of all forests in Northern Eurasia were underlain by permafrost (Supplementary Fig. S4).” This figure only shows permafrost distribution, not forest distribution. Could you overlay the forest distribution on the permafrost map so that the reader can see the distribution of that 29%?

Response 13: In Figure 1, we also show the distribution of all young and old forests across the region plus the distribution of young and old forests underlain by permafrost (i.e. “permafrost forests) during the year 2000.

Comment 14: How is fire represented? This is likely to have large effects on forest age structure under the different climate scenarios. I understand that it is not included as a dynamic process, but do you change the age structure of the forests under the different scenarios to reflect more frequent disturbances? I see from the caveats at the end of the paper that this is not included.

Response 14: The effects of wildfire, either directly or indirectly, are not represented in our simulation analysis.

Comment 15: Fire should have similar impacts to timber harvest in some of the regions. Understory fire versus stand-replacing?

Response 15: In lines 402-406 of the Discussion, we indicate that the responses of forest carbon and nitrogen dynamics recovering from a wildfire are similar to those associated with a clear-cut timber harvest. In lines 408-414, we indicate that permafrost degradation may cause more severe fires (i.e. higher carbon losses) in the future, but may also speed up recovery of the forest from the fire disturbance (which is similar to our analysis of the recovery from timber harvest). However, we also indicate this similarity may not occur if the frequency and severity of wildfires cause an area to be unsuitable for forest growth (lines 414-417), which has already occurred for tens of millions of forested hectares in the Russian Far East.

Comment 16: What are the relative impacts of temperature and moisture on forest growth? How does the model represent drought index under the different climate scenarios? Seems like this is key to predicting forest growth. What is the relative important of temperature versus precipitation differ across regions and climate scenarios? How does the ‘wetness’ of these climate scenarios compare to other modeling efforts?

Response 16: While we agree that it would be interesting to examine how the relative effects of temperature and precipitation on nitrogen availability and carbon sequestration in Northern Eurasia vary over space and time, we think that these issues are secondary to the main issues we

are attempting to address in the reorganized manuscript: the effects of permafrost degradation, land-use legacies and atmospheric N deposition on nitrogen availability and carbon sequestration in Northern Eurasian forests. A follow-up study to examine these temperature and moisture issues in more depth would probably be more useful than attempting to address these issues in this paper.

Comment 17: “Over the 21st century, we project Northern Eurasian ecosystems will sequester 40.6 Pg C under the RCP8.5 scenario and 53.7 Pg C under the RCP4.5 scenario.” So this is about 0.4 and 0.5 Pg C yr⁻¹, which is within the range of estimates for current rates. Why is this exciting? Is the excitement in the spatial distribution of the sinks and sources?

Response 17: With the quantification of the effects of permafrost degradation, land-use legacy and atmospheric N deposition on forest carbon sequestration, we now find that the benefits of nitrogen subsidies from these factors on carbon sequestration by vegetation is somewhat compensated by enhanced losses of carbon from soil organic matter from permafrost degradation such that future rates of carbon sequestration are within the range of estimates for current rates. As a result, other factors may be more responsible for any changes in carbon sequestration that may occur in the region. These insights are described in lines 377-387.

The “excitement” about the spatial distribution of sinks and sources is that the carbon sources in central Siberia during the first half of the 21st century turn into carbon sinks during the second half of the 21st century whereas the carbon sinks in eastern Europe/western Russia are sinks throughout the 21st century (see Fig. 8a). Timber harvest occurs in both regions, but the change in carbon source/sink activity in central Siberia is a result of changes in nitrogen availability caused by climate change and nitrogen subsidies from permafrost degradation (see Fig. 9). These dynamics are now described in lines 313-356.

Comment 18: “Under the RCP8.5 scenario, increases in carbon sequestration rates (0.0040 ± 0.0004 Pg C yr⁻², $r^2 = 0.47$, $p < 0.0001$, $n = 100$) of the undisturbed old forests are supported by concurrent warming-induced increases in net N mineralization (0.065 ± 0.011 Tg N yr⁻², $r^2 = 0.27$, $p < 0.0001$, $n = 100$) throughout the 21st century.” Not clear to me why this correlation is meaningful—could be driven by model structure. Please explain the independence of these estimates and why this might be representative of something other than model structure.

Response 18: The purpose of these regression analyses is to indicate that there is a synchronization of increases in nitrogen availability, represented by net nitrogen mineralization, and increases in carbon sequestration. While there was synchrony between increases in nitrogen availability and increases in carbon sequestration for old forests and secondary forests recovering from timber harvest in both scenarios, there was no synchrony between increases nitrogen availability and carbon sequestration in secondary forests growing on abandoned agricultural land. To better indicate this intent, we added the following sentence “Land-use history affects

both the sensitivity of forests to warming and the synchronization of N availability to carbon sequestration in these forests” in lines 232-233 to introduce this section of the Results.

Page 6, Second paragraph.

Comment 19: Changes to carbon sequestration in permafrost forests gets a fair amount of attention in the main body of the manuscript, yet there are no primary figures that provide direct support for changes in these forests. They are buried in the supplementary information.

Response 19: The distribution of permafrost forests is now shown in Figure 1. The relative importance of trends in net N mineralization and carbon sequestration in permafrost forests compared to all forests is shown in Figure 4. A comparison of Figure 7 to Figure 1 indicates the distribution of changes in timber harvest in permafrost forests and a comparison of Figure 8 to Figure 1 indicates the distribution of changes in carbon sequestration and net N mineralization in permafrost forests. Finally, Figure 9 describes how the response of carbon and nitrogen dynamics to timber harvest in a permafrost forest changes over the 21st century.

Comment 20: How important are these values relative to regional estimates?

Response 20: The changing responses of permafrost forests to warming and land-use change over the 21st century cause these permafrost forests to account for a larger and increasing share of forest carbon sequestration: 10% of forest carbon sequestration at the beginning of the century, 19-24% of forest carbon sequestration at the end of the 21st century. This is indicated in lines 352-356 and shown in Figure 4.

Comment 21: How novel are these mechanisms or conclusions? How does this knowledge change our understanding?

Response 21: We are unaware of any other study that has attempted to look at how nitrogen subsidies from permafrost degradation and land-use legacies may influence future regional carbon sequestration in Northern Eurasia. Our analysis provides new insights into how access to “new” nitrogen resources from permafrost degradation and abandonment of agricultural land may influence regional carbon sequestration in the future. These insights are summarized in lines 359-381 of the Discussion.

Comment 22: How useful is modeling of forest carbon balance that does not include fire, or pests and diseases? How useful is modeling of permafrost carbon balance that does not include carbon stocks below 2 m depth?

Response 22: In our analyses within the reorganized manuscript, we focus on how nitrogen subsidies from permafrost degradation, land-use legacies, and atmospheric N deposition influence carbon sequestration within the context of changes in other factors such as climate, atmospheric chemistry, and land use change. Under these conditions, we have shown that both permafrost degradation and land-use legacies influence how carbon source/sink activity evolves over time. Forests tend to act more as a source of carbon during the early part of the century when little additional nitrogen is available from permafrost degradation and land-use legacies, but tend to act more as a carbon sink later in the century when more nitrogen is available from these factors (lines 204-220). While these three factors clearly influence C/N interactions and C sequestration in the region, we recognize that difficult-to-predict (due to lack of mechanistic understanding) factors such as disturbances from wildfires, pests and diseases, and the decomposition of deep soil organic carbon may increase the total amount of carbon lost from these forests and/or diminish the total amount of carbon sequestered by vegetation. As we now point out in the revised text (lines 388-417, lines 431-439), the carbon sequestration dynamics in these disturbed forests will still be evolving over the 21st century as a result of the nitrogen subsidies associated with permafrost degradation and land-use legacies. Thus, our study provides a solid foundation for follow-up studies to better assess the additional impact of wildfire, pests and diseases and the decomposition of deep soil organic carbon (along with other disturbances and mechanisms) on forest carbon sequestration in a changing world as our understanding of these factors improves over time.

Comment 23: Figure 1 caption: loss should be lost, or alternatively, gained should be gain to match tenses. I find the comparison between the nitrogen circles and the carbon bars difficult to interpret. There is a lot of information contained in this figure, but not much of it seems salient to the text. I would like to see the relative partitioning of regional carbon fluxes in one figure or table rather than having them spread out through 9 figures. I also find that the nitrogen circles are hard to examine. I appreciate the idea that nitrogen cycles, but I would like to see something more quantitative, where I could easily extrapolate data to compare the absolute and relative magnitude of effects. At the very least, I would like to see estimates of uncertainty on these figures.

Response 23: In the Figure caption (now Figure 3), “loss” has been changed to “lost”. The main purposes of this figure are to show: 1) the general dominance of forests over other non-forest ecosystems on regional carbon sequestration; 2) the relative importance of young forests and old forests are different between the two global change scenarios; and 3) nitrogen availability is dominated by net N mineralization. We agree that the figure contained a lot of information that was hard to wade through to see these three main points. To focus the reader on the more important messages from this figure, we combined the results for all Non-forest ecosystems into a single graph and associated nitrogen circles. We also agree that the information in the nitrogen circles were a little hard to decipher so we changed the color scheme to be something easier to discern. As indicated in the Figure legend, the values behind these figures are provided in

Supplementary Tables S5-S12 so a reader does have access to the information behind the graphic to conduct more quantitative analyses.

Comment 24: Figure 2. Estimates of uncertainty? If fire were included in these temporal trends, how might they be different? My guess is that the permafrost forests—largely fire-prone—would not show high forest sequestration.

Response 24: The main purpose of this figure (now Figure 4) is to show that forests dominate the region's carbon sequestration and nitrogen availability (via net N mineralization) trends throughout the 21st century to help justify why we focus our attention of permafrost degradation effects and land-use legacy effects on net N mineralization and carbon sequestration in forests. A secondary purpose is to show that permafrost forests have the potential to become more important carbon sinks in the region from warming and land-use change as described in lines 352-356. Because the focus of our study is the influence of permafrost degradation and land-use legacies on nitrogen availability and carbon sequestration, the potential effects of wildfire on the uncertainty of our estimates is beyond the scope of our reorganized study for this graphic so we did not modify the figure. In lines 402-417 of the Discussion, we indicate that although the fire disturbance itself may release carbon to reduce carbon sequestration, forest regrowth during recovery enhances carbon sequestration later during the century and may benefit from the enhanced nitrogen availability associated with permafrost degradation. Thus, it is not clear that consideration of fires would diminish the positive trend of carbon sequestration in permafrost forest over the 21st century as suggested by the reviewer.

Comment 25: Figure 4. If fire—both understory and stand-replacing—were included in this figure, how might it be different?

Response 25: As Figure 4 (now Figure 7) shows the distribution of different prescribed land uses in Northern Eurasia, the inclusion of fire would have no effect. However, we assume the reviewer meant how might the inclusion of fire change the distribution of carbon sequestration and net N mineralization shown in Figure 5 (now Figure 8). As the focus of our study is the influence of permafrost degradation and land-use legacies on nitrogen availability and carbon sequestration, the potential effects of wildfire on the distribution of net N mineralization and carbon sequestration is beyond the scope of our reorganized study and we did not modify this figure.

Reviewer #2 (Remarks to the Author):

The manuscript by Kicklighter et al shows projections of carbon balance for the northern Eurasian region over the 21st century using the TEM ecosystem model. The model has been

well-tested in high latitude ecosystems, with an interesting representation of how permafrost, nitrogen, and land management dynamics all interact with the carbon cycle. The interesting result here is that nitrogen, land management, and recovery from past disturbances play a relatively large part in the carbon dynamics of the region. Typically people tend to focus on the carbon stored in permafrost soils as the dominant driver of carbon change in the region, but the simulations suggest that, at least over the period of interest here, nitrogen plays a key role and the land use dynamics are also crucial.

My main concern here is on the interpretation of the results: how do the authors know that it is nitrogen from warming soils, rather than direct effects of warming on longer growing seasons, that drives the carbon uptake? The same argument applies to land-use: how do they point to past fertilizer use on abandoned lands, rather than the land abandonment itself, as the driver of carbon uptake? The figures are making a correlative case for this, but the great strength of models like this is that you can unambiguously attribute the drivers of a given change by comparing the response of the system to various forcing scenarios. So to really argue for the importance of nitrogen, it seems like you would need to do an experiment where you hold nitrogen mineralization fixed and allow everything else to change; with that you could identify the role of nitrogen within the suite of other drivers. I don't see any attempt made to do this kind of experiment. Thus the conclusions are somewhat speculative.

Otherwise the paper is interesting and I see it as a worthwhile contribution to the literature of carbon cycle responses to global change at high latitudes.

One final issue is that the data availability statement is weak: it is (like it or not) 2018 and I think the expectation should be that one archives one's results publicly at the time of publication, rather than simply stating that they are available upon reasonable request.

We see three issues to be addressed from the above comments:

Comment 26: How do the authors know that it is nitrogen from warming soils, rather than direct effects of warming on longer growing seasons, that drives the carbon uptake?

Response 26: This was a very good question. Because net N mineralization is influenced by carbon-nitrogen interactions in TEM, there is not an easy way to hold net N mineralization constant to address the reviewer's concern. However, we felt that the more useful question here was how does warming-induced permafrost degradation influence nitrogen availability to affect carbon sequestration. In TEM, the amount of soil organic matter exposed to decomposition is determined by the depth of the simulated active layer. To examine how permafrost degradation might influence nitrogen availability and carbon sequestration, we modified TEM such that the seasonal active layer depth remained constant after the year 2000 so that no additional soil

organic matter would be exposed to decomposition from permafrost degradation and reran the model using the two global scenarios. We then subtracted the results from the “constant active layer depth” simulation from the corresponding results in the simulations where active layer depth was allowed to vary to determine the effect of permafrost degradation. This approach is described in the Methods (lines 729-739) with the results summarized in a new Table 1 and the results of the constant active layer depth simulations provided in Supplementary Tables S35-S38.

Comment 27: How do the authors know that past fertilizer use on abandoned lands, rather than the land abandonment itself, is the driver of carbon uptake?

Response 27: This also was a very good question. To address this issue, we conducted a simulation experiment in which we ran first simulations assuming that no fertilizers were ever applied to croplands with the global change scenarios and then ran a second set of simulations where we assumed that fertilizers were applied to croplands until the year 2000, but then no fertilizers were applied after 2000. Comparison of our “No Fertilizer” simulations to our regular “Fertilized” simulations would provide an estimate of the impact of legacy fertilizer use on forests growing on abandoned croplands. Comparison of the “Pre-2001 Fertilizers” would provide an estimate of the impact of legacy fertilizers for fertilizers applied in the 21st century and could be influenced by future land management of cropland decisions. A comparison of the “Pre-2001 Fertilizer” simulations to the “No Fertilizer” simulations would provide an estimate of the impact legacy fertilizers that had been applied before the 21st century and would not be affected by future land management of cropland decisions. This approach is described in the Methods (lines 740-755) with the results summarized in a new Table 1 and the results of the “no fertilizer” and “pre-2001 fertilizer” simulations provided in Supplementary Tables S39-S50. We found that fertilizer use accounted for about 10.8% of the carbon sequestered by forest vegetation growing on abandoned croplands, but this accounted for only 2.9% of the carbon sequestered by all forest vegetation. Land abandonment itself was the primary driver of carbon uptake associated with the land-use legacies.

Comment 28: Data availability statement is weak.

Response 28: While the data are currently still only available upon reasonable request, we are in the process of arranging to place this data in an archive at DSpace@MIT and plan for this archive to be publicly available before the manuscript is accepted. We set up a similar arrangement for a recent paper in Nature Communications by some of the same co-authors (see <http://hdl.handle.net/1721.1/113296> as an example of this archive).

Reviewer #3 (Remarks to the Author):

Summary: Kicklighter et al. use the TEM model of coupled terrestrial C-N cycling to predict rates of terrestrial C storage across northern Eurasia of future two contrasting scenarios (RCP 4.5 and 8.5) for changes in climate, atmospheric chemistry (CO₂, N deposition) and land use over the next century. They predict that warming-driven increase in soil N mineralization will stimulate net carbon uptake by the region's forests, with additional growth driven by assumed fertilization of croplands later abandoned.

General comments: The paper is clear and easy to read. These simulations appear to be fairly straightforward model projections of regional (N. Eurasia) carbon balance in response to IPCC-projected scenarios, using an established biogeochemistry model. As presented, this model seems structured to yield results generally similar to most other terrestrial ecosystem and earth system models applied to examine terrestrial C storage in response to this suite of projected environmental changes.

For example, TEM and other models structured with classic first-order soil decomposition dynamics generally predict that warming will increase soil N availability to plants that will offset the C losses from soils (e.g., Bonan 2008 Nature Geoscience); it's not wholly apparent if or how this model application yields results that are especially unusual or exceptionally large, or marks a substantial advance in modeling approaches. Other high-profile advances in soil C-N modeling are exploring the role of microbial dynamics with priming or mycorrhizal allocations to enhance plant acquisition of soil N (e.g., Sulman et al. 2014, Nature Climate Change, Wieder et al. 2018 Global Change Biology), a set of processes that are likely to be very important in supplying N to Eurasia's boreal forests, but not yet included in this model analysis.

If this analysis aimed to highlight inclusion of its permafrost model as a novel module, that contrast over past work by this or other models should receive greater emphasis – e.g., contrasting results with and without that module, and comparisons with observations to confirm model performance or identify areas of continued challenge. The role of N from fertilization of cropland later abandoned to forest is interesting and potentially novel, but seems to be largely assumed input of N fertilizer rather than values supported by fertilization data; additional support for that assumption would bolster the model's corresponding conclusions about the role of that N.

Overall, this manuscript would also generally benefit by the addition of some comparisons with observations of important C and/or N cycle pools or fluxes, both to demonstrate model performance under current or altered (experimental) conditions and to yield other insights for model improvement in the future.

We see two issues to be addressed from the above comments:

Comment 29: How are the model results especially unusual or exceptionally large? Or, how does this study mark a substantial advance in modeling approaches?

Response 29: Based on the reviewer's comments, we felt that an analysis that quantified the effects of permafrost degradation and land-use legacies on nitrogen availability and carbon sequestration in forests would provide useful insights that are not currently available. As described in **Responses 2, 6, 26, and 27**, we conducted a number of additional model simulations to quantify the nitrogen subsidies provided by permafrost degradation, land-use legacies, and atmospheric N deposition and their effects on forest carbon sequestration. The approach to quantify the permafrost degradation effects is described in the Methods (lines 729-739) with the results summarized in a new Table 1 and the results of the constant active layer depth simulations provided in Supplementary Tables S35-S38. The approach to quantify fertilizer application effects is described in the Methods (lines 740-755) with the results summarized in a new Table 1 and the results of the "no fertilizer" and "pre-2001 fertilizer" simulations provided in Supplementary Tables S39-S50.

In addition, we indicated the potential limitation of our model analysis by not considering the influence of priming and mineral protection of SOM on decomposition and nitrogen availability, and how our model estimates might change if we did consider these mechanisms (lines 445-453). Unfortunately, the Wieder et al. 2018 Global Change Biology paper does not appear to be publicly available yet so I was unable to incorporate information from that paper into our analysis.

Comment 30: The manuscript would generally benefit by the addition of some comparisons with observations of important C and/or N cycle pools or fluxes, both to demonstrate model performance under current or altered (experimental) conditions and to yield other insights for model improvement in the future.

Response 30: As described in **Response 3**, we added a new subsection to the Results, "Current carbon stocks and carbon sinks in Northern Eurasia" (lines 63-88) and new Supplementary Tables S2 and S3 where we compare our regional estimates of carbon stocks in vegetation and soils to inventory estimates by other studies. While our estimates of soil organic carbon compared fairly well with inventory estimates, our estimates of vegetation carbon are higher than the inventory estimates. We think this overestimate is a result of not considering the impacts of wildfire on vegetation during our simulations and perhaps issues with the ability of our spatially explicit time series data set to capture historical land-use change realistically. In addition, we found our simulated estimate of fertilizer applications to croplands during the 1990s corresponded well to inventory estimates of fertilizer data for the region during the same period.

Reviewers' comments:

Reviewer #2 (Remarks to the Author):

The revised paper addresses well the comments of the reviewers; I feel it is ready for publication.

Reviewer #4 (Remarks to the Author):

The manuscript quantifies nitrogen subsidies from permafrost degradation and land-use legacies on land carbon sequestration in Northern Eurasia during the 21st century using a process based model, TEM. I found the work to be very timely and exceptional.

I also really liked the simulation experiments conducted by the authors through TEM.

The reviewers can however significantly improve the manuscript further. My major comments are described below:

First, the manuscript can be made concise at certain places, but the organization can be definitely improved. One thing that bothered me was that the authors had a haphazard description of the role that permafrost degradation and abandoned croplands can play in future trajectories. I think each of these roles needs to be described individually first, and then the emphasis should be made on why a process-based model is necessary to tweak out the interrelationships among all of these factors. This did not come across in the intro section.

Second, the definition of nitrogen legacies as used by the authors is buried in the manuscript and gets described much later. This needs to be brought up.

Third, a big impact statement was missing from the manuscript. For example, how important are changes in Northern Eurasia on a global scale. Or else, where are the hot spots located in this region, that are very important to monitor for future land management plans?

Response to Reviewers' Comments

Reviewer #2 (Remarks to the Author):

Comment 5: The revised paper addresses well the comments of the reviewers; I feel it is ready for publication.

Response 5: We thank the reviewer for her/his comments and assume that no additional revisions to the manuscript are needed to satisfy the reviewer.

Reviewer #4 (Remarks to the Author):

Comment 6: The manuscript quantifies nitrogen subsidies from permafrost degradation and land-use legacies on land carbon sequestration in Northern Eurasia during the 21st century using a process based model, TEM. I found the work to be very timely and exceptional. I also really liked the simulation experiments conducted by the authors through TEM. The reviewers can however significantly improve the manuscript further. My major comments are described below:

First, the manuscript can be made concise at certain places, but the organization can be definitely improved. One thing that bothered me was that the authors had a haphazard description of the role that permafrost degradation and abandoned croplands can play in future trajectories. I think each of these roles needs to be described individually first, and then the emphasis should be made on why a process-based model is necessary to tweak out the interrelationships among all of these factors. This did not come across in the intro section.

Response 6: To improve the description of the role that permafrost degradation and abandoned croplands can play in future trajectories and why a process-based model is necessary to tweak out the interrelationships, we reorganized and modified the Introduction in **lines 27-76**:

“The availability of soil inorganic nitrogen (N) is a critical controller of plant productivity and carbon (C) sequestration in many temperate and boreal ecosystems¹⁻⁴ including those in Northern Eurasia⁵. Human activities have directly altered N availability in these ecosystems by providing N subsidies associated with enhanced atmospheric N deposition from fossil fuel combustion, and the application of N fertilizers to croplands⁶. These activities have also altered environmental conditions, including climate, to indirectly affect the metabolic rates of biological processes, such as microbially-mediated transformations of organic N compounds to inorganic N (i.e., N mineralization) associated with the decomposition of soil organic matter (SOM) and biological N fixation, to potentially increase N availability⁶⁻¹². Warming-induced permafrost degradation also provides an additional N subsidy to vegetation.

Historically, in high latitude ecosystems such as those in Northern Eurasia, permafrost has protected some ancient SOM from decomposition and the associated cool soil thermal regimes have led to slow decay rates of the more recent SOM to limit N availability to vegetation⁹⁻¹¹. With warming, more soil N may become available to vegetation, if favorable moisture conditions exist, as SOM decay rates increase in the thawed soil layers⁹⁻¹². In addition, permafrost degradation will expose some protected SOM to decomposition^{13,14} and increase N mineralization to provide an additional “recycled” N subsidy to vegetation that is not currently available. Because this recycled N subsidy is associated with a concurrent loss of C from the enhanced decomposition of SOM, the potential effect of this enhanced N availability on net land C sequestration will depend on the type of vegetation cover. As the carbon-to-nitrogen ratio (C:N) of wood is an order of magnitude greater than the C:N of SOM¹⁵, trees may be able to sequester more atmospheric carbon dioxide from the recycled N subsidy than is lost from SOM decomposition associated with permafrost degradation. In contrast, the C:N of non-woody vegetation is more similar to the C:N of SOM such that the benefits of this recycled N subsidy on C sequestration may be more limited.

Besides warming, land-use legacies may also provide a “recycled” N subsidy to influence future land C sequestration dynamics. The abandonment of agricultural land (i.e. croplands and pastures) allows access to N sources not currently available to the subsequent natural ecosystem. This N subsidy to natural ecosystems is balanced by the exact corresponding loss of N from agricultural lands at the regional scale and changes over time based on the area of agricultural land abandoned. As a legacy of past land management, this N subsidy may be enhanced from past fertilizer applications to croplands^{16,17}. Although fertilizers are applied to croplands to enhance crop yield, some of the fertilizer N will remain as part of the crop residues that eventually become part of the SOM. With the abandonment of croplands, decomposition and mineralization of this “legacy” fertilizer-enhanced SOM can then increase N availability to the subsequent natural vegetation to influence land C sequestration. Similar to permafrost degradation, the influence of this legacy fertilizer N subsidy on land C sink/source dynamics will depend on the cover type of the secondary vegetation.

Because the N subsidies from permafrost degradation and land-use legacies depend on the recycling of N from SOM, the potential benefits of these N subsidies need to be evaluated within the context of other environmental conditions that affect SOM decomposition and mineralization, or that provide other N subsidies such as atmospheric N deposition¹⁸⁻²². In addition, the benefits of these recycled N subsidies to future land C sequestration will lag an initial loss of C associated with the enhanced decomposition associated with exposure of new SOM. This suggests that the response of land C sequestration to climate and land-use change may evolve over time in Northern Eurasian ecosystems. While previous modeling studies²³⁻²⁶ have shown the importance of carbon-nitrogen interactions on the overall response of land C sequestration in pan-arctic ecosystems to future warming, they have not examined how changes in climate, land use, and atmospheric chemistry may interact to affect the influence of future N availability on C sequestration in these ecosystems.”

***Comment 7:** Second, the definition of nitrogen legacies as used by the authors is buried in the manuscript and gets described much later. This needs to be brought up.*

Response 7: As part of the reorganization of the Introduction, we brought up text that occurred later in the manuscript and expanded this text to define nitrogen legacies as we use the term (see **lines 52-62**):

“Besides warming, land-use legacies may also provide a “recycled” N subsidy to influence future land C sequestration dynamics. The abandonment of agricultural land (i.e. croplands and pastures) allows access to N sources not currently available to the subsequent natural ecosystem. This N subsidy to natural ecosystems is balanced by the exact corresponding loss of N from agricultural lands at the regional scale and changes over time based on the area of agricultural land abandoned. As a legacy of past land management, this N subsidy may be enhanced from past fertilizer applications to croplands^{16,17}. Although fertilizers are applied to croplands to enhance crop yield, some of the fertilizer N will remain as part of the crop residues that eventually become part of the SOM. With the abandonment of croplands, decomposition and mineralization of this “legacy” fertilizer-enhanced SOM can then increase N availability to the subsequent natural vegetation to influence land C sequestration.”

***Comment 8:** Third, a big impact statement was missing from the manuscript. For example, how important are changes in Northern Eurasia on a global scale. Or else, where are the hot spots located in this region, that are very important to monitor for future land management plans?*

Response 8: To indicate how changes in nitrogen availability from permafrost degradation, legacy soil nitrogen from abandoned agricultural land, and atmospheric nitrogen deposition may affect future land management plans for sequestering carbon in land ecosystems, we modified the Abstract (**lines 14-24**) and added a paragraph (**lines 87-101**) to the end of the Introduction as indicated in **Response 3** above. In addition, we modified or added the following text to the Discussion (**lines 417-431**):

“As a result, we estimate that C sequestration in Northern Eurasia during the 21st century, on average, will likely be similar to current rates of C sequestration. These average rates, however, conceal the influence of important changes in temporal trends and spatial patterns of C sequestration and loss occurring in the region.

The asynchronous timing of C gain by trees and C loss from soils associated with the N subsidies causes the size and geographical distribution of important regional C sources and sinks to evolve over time. Although permafrost degradation tends to diminish C sequestration in forests overall during the 21st century, N subsidies from permafrost degradation help Siberian forests recover from timber harvest more rapidly such that these forests become larger C sinks rather than C sources during the latter part of the century. In contrast, N subsidies from the

abandonment of agricultural land, particularly in the western part of the region, during the first half of the 21st century, tend to enhance C sequestration in these forests throughout the 21st century, but the relative benefits of these enhanced C sinks diminish over time as these secondary forests regrow. As a result, these secondary forests become a less important component of the regional C sink during the latter part of the 21st century.”

Finally, we modified or added the following text to the end of the Discussion (**lines 510-514**):

“Regardless, our study suggests that permafrost degradation and land management decisions will have a large influence on N availability to affect how land C dynamics in Northern Eurasia will evolve in response to future changes in climate, atmospheric chemistry, and disturbances. Thus, carbon-nitrogen interactions need to be considered when assessing sub-regional and regional impacts of global change policies.”

REVIEWERS' COMMENTS:

Reviewer #4 (Remarks to the Author):

The reviewers have fairly addressed all my comments.

Responses to Reviewer's Comments

REVIEWERS' COMMENTS:

Reviewer #4 (Remarks to the Author):

Comment 16: The reviewers have fairly addressed all my comments.

Response 16: We assumed the reviewer meant that the “authors” have fairly addressed all my comments rather than “reviewers”. Thus, we assume that the reviewer is satisfied with the previously revised manuscript. As there are no other referee comments, all of our final revisions to the manuscript are to made to address the journal’s style and format requirements.